# Haplotype-based analysis distinguishes maternal-fetal genetic contribution to pregnancy-related outcomes

**Amit K. Srivastava**[1], **Julius Juodakis**[2], **Pol Sole-Navais**[2], **Jing Chen**[3], **Jonas Bacelis**[2,4], **Kari Teramo**[5†], **Mikko Hallman**[6], **Pal R. Njølstad**[7,8,9], **David M. Evans**[10,11], **Bo Jacobsson**[12,13], **Louis J. Muglia**[1,14], **Ge Zhang**[1,14]*

1 Division of Human Genetics, Center for Prevention of Preterm Birth, Perinatal Institute and March of Dimes Prematurity Research Center Ohio Collaborative, Cincinnati Children's Hospital Medical Center, Cincinnati, Ohio, United States of America, 2 Department of Obstetrics and Gynecology, Institute of Clinical Sciences, Sahlgrenska Academy, University of Gothenburg, Gothenburg, Sweden, 3 Division of Biomedical Informatics, Cincinnati Children's Hospital Medical Center, Cincinnati, Ohio, United States of America, 4 Region Västra Götaland, Sahlgrenska University Hospital, Department of Obstetrics and Gynecology, Gothenburg, Sweden, 5 Obstetrics and Gynecology, University of Helsinki and Helsinki University Hospital, Helsinki, Finland, 6 PEDEGO Research Unit and Medical Research Center Oulu, University of Oulu and Department of Children and Adolescents, Oulu University Hospital, Oulu, Finland, 7 KG Jebsen Center for Diabetes Research, Department of Clinical Science, University of Bergen, Bergen, Norway, 8 Division of Health Data and Digitalization, Department of Genetics and Bioinformatics, Norwegian Institute of Public Health, Oslo, Norway, 9 Center for Medical Genetics and Molecular Medicine, Haukeland University Hospital, Bergen, Norway, 10 Institute for Molecular Bioscience, Frazer Institute, The University of Queensland, Brisbane, Australia, 11 Medical Research Council Integrative Epidemiology Unit, University of Bristol, Bristol, United Kingdom, 12 Department of Obstetrics and Gynecology, Sahlgrenska Academy, University of Gothenburg, Gothenburg, Sweden, 13 Department of Genetics and Bioinformatics, Area of Health Data and Digitalization, Norwegian Institute of Public Health, Oslo, Norway, 14 Department of Pediatrics, University of Cincinnati College of Medicine, Cincinnati, Ohio, United States of America

† Deceased

* Ge.Zhang@cchmc.org

## Abstract

Genotype-based approaches for the estimation of SNP-based narrow-sense heritability ($\hat{h}^2$) have limited utility in pregnancy-related outcomes due to confounding by the shared alleles between mother and child. Here, we propose a haplotype-based approach to estimate the genetic variance attributable to three haplotypes - maternal transmitted ($\hat{h}_{m1}^2$), maternal non-transmitted ($\hat{h}_{m2}^2$) and paternal transmitted ($\hat{h}_{p1}^2$) in mother-child pairs. We show through extensive simulations that our haplotype-based approach outperforms the conventional and contemporary approaches for resolving the contribution of maternal and fetal effects, particularly when m1 and p1 have different effects in the offspring. We apply this approach to estimate the explicit and relative maternal-fetal genetic contribution to the phenotypic variance of gestational duration and gestational duration-adjusted fetal size measurements at birth in 10,375 mother-child pairs. The results reveal that variance of gestational duration is mainly attributable to m1 and m2 ($\hat{h}_{m1}^2 = 17.3\%$, S.E. $= 5.2\%$; $\hat{h}_{m2}^2 = 12.2\%$, S.E. $= 5.2\%$; $\hat{h}_{p1}^2 = 0.0\%$, S.E. $= 5.0\%$). In contrast, variance of fetal size measurements at birth are mainly attributable to m1 and p1 ($\hat{h}_{m1}^2 = 18.6 - 36.4\%, \hat{h}_{m2}^2 = 0.0 - 5.2\%$ and $\hat{h}_{p1}^2 = 4.4 - 13.6\%$). Our results suggest that gestational duration and fetal size measurements are primarily genetically determined

**Data availability statement:** The data under-lying this article cannot be shared publicly to protect the interest and privacy of individuals who participated in the study. However, the individual-level phenotype and genotype data can be accessed by submitting applications to and upon approval by the corresponding entities who are in charge of the distribution of the data sets (e.g., ALSPAC, FIN, MoBa, and dbGaP). This is to ensure that the proposed study aims are consistent with the informed consent under which the data or samples were collected and appropriate data safety and secu-rity measures are in place to protect against data breaches and unauthorized use. ALSPAC data are available to scientists on request to the ALSPAC Executive Committee (ALSPAC-exec@bristol.ac.uk) or via the website (http://www.bristol.ac.uk/alspac/researchers/access/), which also provides full details and distributions of the ALSPAC study variables. The detailed policy of data sharing can be found in the ALSPAC data management plan (http://www.bristol.ac.uk/alspac/researchers/data-access/documents/alspac-data-management-plan.pdf). Access to the FIN data requires approval by our Leadership Committee to ensure appropri-ate use and protection of participant privacy. Researchers interested in using the dataset for bona fide studies can either contact our pro-gram manager Xin Tang at Xin.Tang@cchmc.org or submit the application form at (https://hpg.research.cchmc.org/fin_data.html). MoBa data is available to researchers and research groups at both the Norwegian Institute of Public Health and other research institutions nationally and internationally. The research must adhere to the aims of MoBa and the participants' consent. All use of data and biological material from MoBa is subject to Norwegian legislation. Terms for applying for access to data and links to the application form and information can be found at https://www.fhi.no/en/studies/moba/for-forskere-artikler/research-and-data-access/. Access to the DNBC (phs000103.v1.p1), and HAPO (phs000096.v4.p1) individual-level phenotype and genetic data can be obtained through dbGaP Authorized Access portal (https://dbgap.ncbi.nlm.nih.gov/dbgap/aa/wga.cgi?page=login). The informed consent under which the data or samples were collected is the basis for determining the appropriateness of sharing data through unrestricted-access databases or NIH-designated controlled-access data repositories. Example scripts, associated binaries and instructions for applying the approach can be found here: https://github.com/amitsrivastava-cchmc/H-GCTA.

by the maternal and fetal genomes, respectively. In addition, a greater contribution of m1 as compared to m2 and p1 ($\hat{h}_{m1}^2 - \hat{h}_{m2}^2 - \hat{h}_{p1}^2 > 0$) to birth length and head circum-ference suggests a substantial influence of correlated maternal-fetal genetic effects on these traits. Our newly developed approach provides a direct and robust alternative for resolving explicit maternal and fetal genetic contributions to the phenotypic variance of pregnancy-related outcomes.

## Author summary

Unlike other complex traits, pregnancy-related outcomes are influenced by both the maternal and fetal genotypes. Conventional genotype-based approaches considering individuals as an analytical unit, therefore, suffer from a bias due to confounding of the shared alleles between the mother and child. We present a unique haplotype-based approach considering mother-child pairs as a single analytical unit with maternal trans-mitted (m1), maternal non-transmitted (m2), and paternal transmitted (p1) haplotypes. Maternal transmitted haplotypes influence pregnancy-related outcomes through both the mother and child whereas maternal non-transmitted and paternal transmitted haplo-types influence pregnancy-related outcomes only through mother and child, respectively. Using extensive simulations, we show that our haplotype-based approach outperforms the conventional GCTA and contemporary M-GCTA approach for resolving maternal and fetal genetic contributions to pregnancy-related outcomes, particularly in the pres-ence of parent-of-origin effects (POEs). We implement our newly developed approach to estimate the explicit and relative maternal-fetal genetic contributions to the phenotypic variance of gestational duration and gestational duration-adjusted fetal size measure-ments at birth in 10,375 mother-child pairs. Our results reveal that gestational duration and birth weight are primarily influenced by maternal and fetal genomes, respectively whereas birth length and head circumference have substantial influence of correlated maternal-fetal genetic effects or POEs.

## Introduction

Narrow sense heritability ($h^2$) is the proportion of phenotypic variance in a population attribut-able to additive genetic values (breeding values) [1]. Generally, the concept of the $h^2$ estimation comes from balanced designs – regression of a child's phenotype on mid-parent phenotype, correlation of full or half sibs and differences in the correlation of monozygotic and dizygotic twins [1]. However, in a population with mixed relationships, linear mixed model (LMM) is the most flexible approach accounting for both fixed and random effects [1–5].

Over the last decade, various methods [6] including Genome-based Restricted Maximum Likelihood (GREML) [7, 8], Linkage Disequilibrium Adjusted kinships (LDAK) [9], threshold Genomic Relatedness Matrices (Threshold-GRMs) [10], LD Score regression (LDSC) [11] and Phenotype Correlation-Genotype Correlation (PCGC) [12] have been developed to estimate SNP-based narrow-sense heritability (commonly known as SNP-based heritability or SNP-heritability - $\hat{h}^2$) [13]. In addition, variants of these approaches such as GREML-MAF stratified (GREML-MS) [14], GREML-LD and MAF stratified (GREML-LDMS) [15] and LDAK-MAF stratified (LDAK-MS) [16] have enabled partitioning of the genetic variance into additive and non-additive components as well as variance components attributable to chromosomes, genes and inter-genic regions. The above approaches have helped explain a large proportion

**Funding:** This work is supported by grants from the Eunice Kennedy Shriver National Institute of Child Health & Human Development of the National Institutes of Health under Award Number R01HD101669, the Burroughs Wellcome Fund (10172896), the Bill and Melinda Gates Foundation (OPP1175128), the March of Dimes Prematurity Research Center Ohio Collaborative, and the Cincinnati Children's Hospital Medical Center (GAP/RIP). The funders had no role in study design, data collection and analysis, decision to publish, or preparation of the manuscript. A.K.S., P.S.-N., J.C., B.J., and G.Z. received salary support from the National Institutes of Health (NIH). Additionally, A.K.S. and G.Z. received salary support from the Burroughs Wellcome Fund, the Bill and Melinda Gates Foundation, and the March of Dimes. The Norwegian Mother, Father and Child Cohort Study is supported by the Norwegian Ministry of Health and Care Services and the Ministry of Education and Research. This research is part of the HARVEST collaboration, supported by the Research Council of Norway (#229624). The genotyping and analyses were supported by the grants from: Jane and Dan Olsson Foundations (Gothenburg, Sweden), Swedish Medical Research Council (2015-02559), Norwegian Research Council/FUGE (grant no. 151918/S10; FRI-MEDBIO 249779), March of Dimes (21-FY16-121), and the Burroughs Wellcome Fund Preterm Birth Research Grant (10172896) and by Swedish government grants to researchers in the public health sector (ALFGBG-717501, ALFGBG-507701, ALFGBG-426411). The UK Medical Research Council and Wellcome (Grant ref: 102215/2/13/2) and the University of Bristol provide core support for ALSPAC. GWAS data was generated by Sample Logistics and Genotyping Facilities at Wellcome Sanger Institute and LabCorp (Laboratory Corporation of America) using support from 23andMe. A comprehensive list of grants funding is available on the ALSPAC website (http://www.bristol.ac.uk/alspac/external/documents/grant-acknowledgements.pdf). The DNBC datasets used for the analyses described in this manuscript were obtained from dbGaP at http://www.ncbi.nlm.nih.gov/sites/entrez?db=gapthroughdbGaP accession number phs000103.v1.p1. The GWAS of Prematurity and its Complications study is one of the genome-wide association studies funded as part of the Gene Environment Association Studies (GENEVA) under the Genes, Environment and Health Initiative (GEI). The HAPO datasets used for the analyses described in this manuscript were obtained

of the missing heritability in various complex diseases and quantitative traits [8–11,13,16–18]. Nevertheless, conventional approaches utilizing an individual's genotype information are less suited for pregnancy-related outcomes which are jointly influenced by direct fetal and indirect parental genetic effects [19–22]. In recent years, several studies using genotype information in mother-child duos [19,21,23–26] and parent-child trios [27] have examined the contribution of parental genetic effects [28, 29] and fetal genetic effects in various pregnancy-related outcomes. However, these approaches are based on several assumptions including equal effects of maternal and paternal transmitted alleles in child. Hence, the estimation of heritability in pregnancy-related outcomes demands a direct approach with relaxed assumptions.

Here, we consider mother-child pair as a single analytical unit consisting of three haplotypes corresponding to maternal transmitted (m1), maternal non-transmitted (m2) and paternal transmitted (p1) alleles [30–32]. Use of such an analytical unit provides an advantage over conventional approaches based on individual's genotype information by avoiding the confounding of m1 which can influence pregnancy-related outcomes through both the mother and child (Fig 1A) [22]. We generate three separate genetic relatedness matrices M1, M2 and P1 using only m1, only m2 and only p1, respectively. We fit all three matrices simultaneously in a linear mixed model (LMM) to estimate variance attributable to each haplotype (Fig 1B). Although our approach doesn't directly estimate SNP-heritability, we use $\hat{h}_{m1}^2$, $\hat{h}_{m2}^2$ and $\hat{h}_{p1}^2$ to represent variance attributable to m1, m2 and p1 respectively for the comparison purposes. We compare the behavior of our newly developed haplotype-based genome-wide complex trait analysis approach (H-GCTA) with existing genotype-based approaches such as Genome-wide Complex Traits Analysis (GCTA) [7, 8] and Maternal-Genome-wide Complex Traits Analysis (M-GCTA) [21,24] approach using simulated phenotypes with varying contributions and correlations of maternal and fetal genetic effects. We show that H-GCTA outperforms the conventional and other contemporary approaches, particularly when the maternal and paternal transmitted alleles have different effects (e.g., parent-of-origin effects - POEs) on a fetal trait and traits with joint maternal-fetal effects.

We further apply our approach to a cohort of 10,375 mother-child pairs to estimate the explicit and relative contribution of maternal-fetal genetic effects to the phenotypic variance of gestational duration and gestational duration adjusted fetal size measurements at birth, including birth weight, birth length and head circumference. Our results suggest that genetic variance in gestational duration is primarily attributable to the maternal genome, i.e., the maternal transmitted (m1) and non-transmitted (m2) alleles, whereas genetic variance in fetal size measurements at birth are largely attributable to fetal genome, i.e., maternal transmitted (m1) and paternal transmitted (p1) alleles. In addition, a higher attribution to m1 as compared to m2 and p1 ($\hat{h}_{m1}^2 - \hat{h}_{m2}^2 - \hat{h}_{p1}^2 > 0$) suggests a large contribution of correlated maternal-fetal genetic effects to the variance of birth length and head circumference. Our haplotype-based approach provides a direct method with relaxed underlying assumptions to estimate the explicit and relative maternal-fetal contributions to the phenotypic variance of pregnancy-related outcomes.

## Results

### Heritability estimation using simulated data

We first evaluated the utility and robustness of H-GCTA using simulated phenotypes based on the real genotype data from a homogenous cohort (Avon Longitudinal Study of Parents And Children; ALSPAC) with 5,369 mother-child pairs and pooled dataset (diverse European populations, including ALSPAC) with 10,375 mother-child pairs. Traits were simulated with varying contributions and correlation of maternal and fetal genetic effects (Table 1 and methods).

from dbGaP at http://www.ncbi.nlm.nih.gov/
sites/entrez?db=gapthroughdbGaP accession
number phs000096.v4.p1. This study is part
of the Gene Environment Association Studies
initiative (GENEVA) funded by the trans-NIH
Genes, Environment, and Health Initiative (GEI).

**Competing interests:** The authors have
declared that no competing interests exist.

All traits were simulated with a total genetic variance at 50%, using a randomly selected set of 10,000 causal variants. Traits with correlated maternal-fetal genetic effects were simulated using the same set of causal variants in mother and child. In addition, we also incorporated different levels of POEs (maternal and paternal transmitted alleles had different effects in fetus) in varying proportion of causal variants for traits with only fetal and joint maternal-fetal effects (Table 1 and methods).

We compared the performance of H-GCTA with conventional GCTA approach and a contemporary M-GCTA approach for each simulated trait. For each approach, we estimated the genetic variance using three models – GREML, LDAK-Thin (where all pruned SNPs were given equal weights) and LDAK with SNP-specific weights (hereafter referred as LDAK-Weights, where each SNP had different weights based on pair-wise LD) (S1 Fig). Using any particular approach, each model yielded similar results when used with recommended α values (GREML: α = -1.0; LDAK and LDAK-Thin: α = -0.25) which represents the extent to which minor allele frequency (MAF) influences the variance of SNP effects on phenotypes [16]. We observed that the estimated genetic variance was similar in pooled datasets and homogenous cohort ALSPAC. However, due to small sample size, the estimated genetic variance in ALSPAC cohort had larger standard errors (S3 and S4 Figs and S5–S12 Tables). Here, we discuss the results of simulated traits from pooled dataset using GREML (α = -1.0) fitted through GCTA, M-GCTA and H-GCTA approach.

## Heritability of simulated traits with only maternal effects

Using conventional approach for maternal traits in mothers and children separately, the estimated SNP-heritability ($\hat{h}^2$) based on maternal (m) and fetal (f) genotypes was 45.0% (S.E. = 8.6%) and 13.6% (S.E. = 8.6%) respectively (Fig 2A and S13 Table). We also used M-GCTA in mother-child duos to estimate the variance attributable to indirect maternal effect ($\hat{h}^2_{M'} = 40.6\%, \text{S.E.} = 6.1\%$), direct fetal effect ($\hat{h}^2_G = -2.2\%, \text{S.E.} = 6.0\%$) and maternal-fetal covariance ($\hat{h}^2_D = 5.4\%, \text{S.E.} = 5.2\%$) (Fig 2A and S13 Table). Using H-GCTA for maternal traits in mother-child duos, variance attributable to maternal transmitted alleles ($\hat{h}^2_{m1}$), maternal non-transmitted alleles ($\hat{h}^2_{m2}$), and paternal transmitted alleles ($\hat{h}^2_{p1}$) were 25.5% (S.E. = 4.8%), 22.9% (S.E. = 4.5%) and -3.0% (S.E. = 4.2%) respectively (Fig 2A and S13 Table). M-GCTA and H-GCTA accurately distinguished the maternal origin of the simulated traits; however, the conventional GCTA also showed a superficial contribution from the fetal genome (13.6%, approximately one quarter of the $\hat{h}^2$ based on maternal genotype) due to 50% alleles shared between mother and child (S13 Table).

## Heritability of simulated traits with only fetal effects

Like maternal traits, we used conventional GCTA to estimate $\hat{h}^2$ for fetal traits in mothers and children separately. The estimated $\hat{h}^2$ based on m and f were 10.9% (S.E. = 8.8%) and 51.9% (S.E. = 8.8%) respectively (Fig 2B and S14 Table). Similarly, using M-GCTA for fetal traits in mother-child duos, variance attributable to indirect maternal effect (M'), direct fetal effect (G) and direct-indirect effect covariance (D) were -3.0% (S.E. = 6.0%), 52.7% (S.E. = 6.1%) and 0.0% (S.E. = 4.7%) respectively (Fig 2B and S14 Table). Using H-GCTA in mother-child duos, we estimated the variance of the simulated fetal traits attributable to m1 ($\hat{h}^2_{m1} = 29.6\%, \text{S.E.} = 4.3\%$), m2 ($\hat{h}^2_{m2} = -3.6\%, \text{S.E.} = 4.4\%$) and p1 ($\hat{h}^2_{p1} = 24.6\%, \text{S.E.} = 4.4\%$) (Fig 2B and S14 Table). While conventional GCTA estimated superficial contributions from maternal genotypes besides fetal genotypes, M-GCTA and H-GCTA clearly showed the fetal origin of the simulated phenotypes. As compared to M-GCTA, H-GCTA further resolved almost equal contributions from maternal and paternal transmitted alleles through m1 and p1.

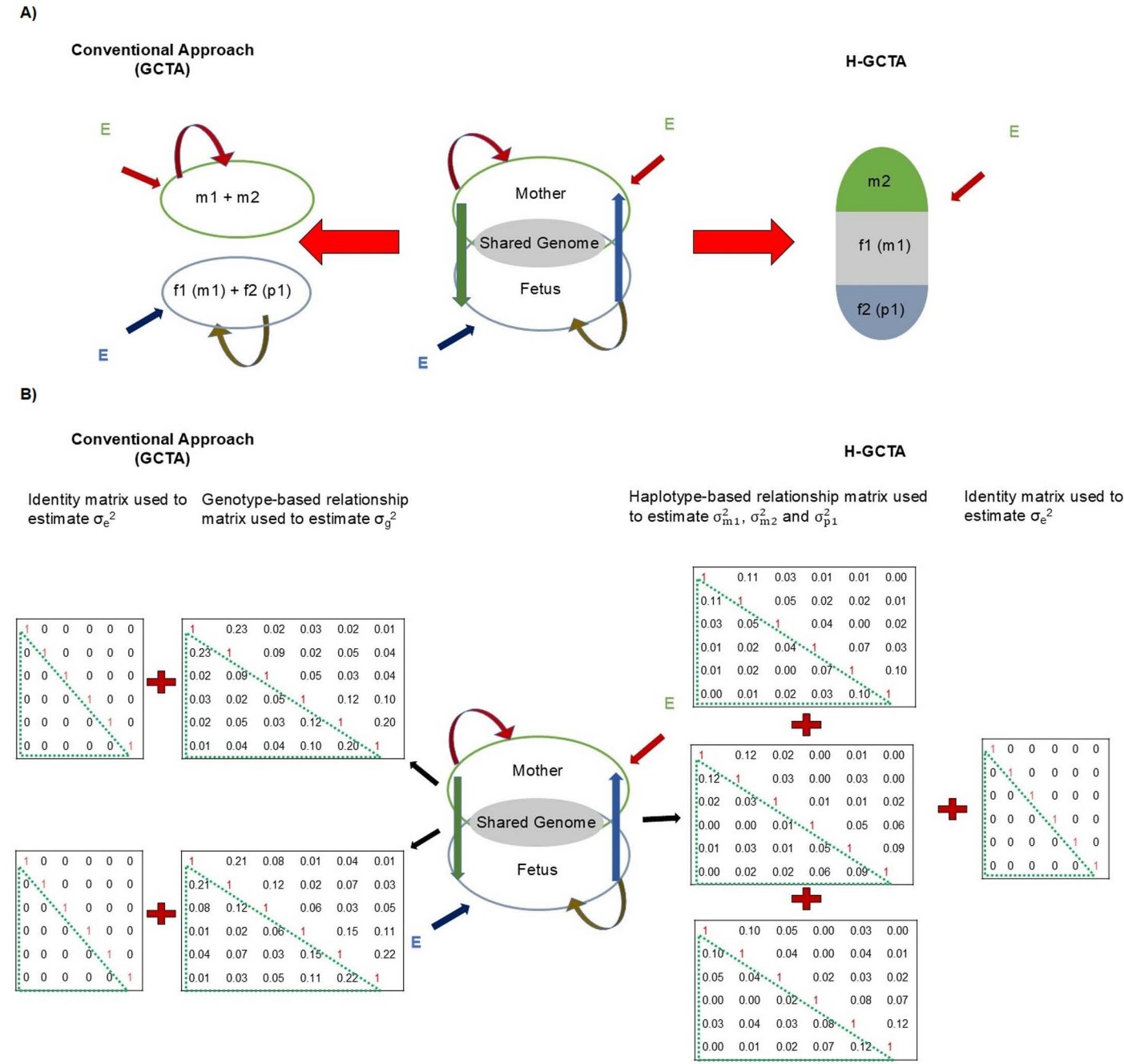

**Fig 1. Comparison of conventional (GCTA) and H-GCTA approach.** A) Schematic representation of the difference between the conventional genotype-based and newly developed haplotype-based analysis approach; the left part of the figure represents the conventional approach based on genotypes of mother and child separately and the right part represents haplotype-based analysis by treating mother/child pairs as analytical units. Green vertical arrow represents maternal genetic effects in fetus, whereas Blue one represents fetal genetic effects in mother during pregnancy. Red and Golden curved arrows represent maternal and fetal genetic effects in mother and fetus, respectively. Red and Indigo slant arrows represent the environmental effects on mother and fetus, respectively. m1 (f1): Maternal transmitted alleles; m2: Maternal non-transmitted alleles; f2 (p1): paternal transmitted alleles; E: Environmental factors. B) Schematic representation of the difference between conventional approach of heritability estimation utilizing genotype-based GRMs and our approach utilizing haplotype-based GRMs (representing the example of mother-child duos). While Conventional GCTA approach fits individual's genotype-based GRM separately in mothers and children (left side), haplotype-based approach fits three haplotype-based GRMs together (right side). $\sigma_g^2$: phenotypic variance attributable to mothers' or children's genotypes; $\sigma_{m1}^2$, $\sigma_{m2}^2$ and $\sigma_{p1}^2$: phenotypic variance attributable to m1, m2 and p1 respectively; $\sigma_e^2$: phenotypic variance attributable to E.

**Table 1. Genetic models for different simulation conditions.**

| Causal effects | Genetic values | Notes |
|---|---|---|
| Only maternal | $(g_{m1} * u_{m1}) + (g_{m2} * u_{m2})$ | $u_{m1} = u_{m2} = u_m$ |
| Only fetal | $(g_{f1} * u_{f1}) + (g_{f2} * u_{f2})$ | $u_{f1} = u_{f2} = u_f$ |
| Joint maternal-fetal* | $(g_{m1} * u_{m1}) + (g_{m2} * u_{m2}) + (g_{f1} * u_{f1}) + (g_{f2} * u_{f2})$ | $u_{m1} = u_{m2} = u_m$; $u_{f1} = u_{f2} = u_f$ with average $cor(u_m, u_f) = \rho$, where $\rho$ = {0.0, -0.5, -1.0, 0.5, 1.0}. For $\rho$ = 0.0, same set and independent sets of causal variants were selected from mothers and fetuses. |
| Only fetal with POEs | $(g_{f1} * u_{f1}) + (g_{f2} * u_{f2})$ | $u_{f1}$ = (1 - I) * $u_{f2}$, where I = {0.25, 0.50, 0.75, 1.0}. POEs were introduced to different proportions of causal variants. |

Genetic models for simulation conditions – simulated maternal traits, fetal traits, traits with joint maternal-fetal effects and fetal traits with different levels of parent-of-origin effects (POEs), incorporated in varying proportion of causal variants. Correlated maternal-fetal effects were randomly drawn from multivariate normal distribution ($u \sim MVN(\begin{smallmatrix} 0 \\ 0 \end{smallmatrix}, \begin{bmatrix} 1 & \rho \\ \rho & 1 \end{bmatrix})$) where, $\rho$ represents correlation of maternal and fetal genetic effects which is equal to covariance of maternal-fetal genetic effects in a standard normal distribution. POEs were incorporated by reducing the effect of maternal transmitted alleles (m1) in comparison to paternal transmitted alleles (p1) by multiplying effects of m1 with (1 – I) where I is the imprinting factor such as 0.25, 0.50, 0.75 and 1.0. In case of I =1.0, m1 has no effect which represents complete imprinting whereas other I values represent partial imprinting. u represents the Allelic effects; $g_{m1}$ and $g_{m2}$ represent the vectors corresponding to maternal transmitted and non-transmitted alleles in mother whereas $g_{f1}$ and $g_{f2}$ represent the vectors corresponding to maternal and paternal transmitted alleles in child, respectively. It is noteworthy that $g_{m1}$ and $g_{f1}$ are same in phased mother-child pairs. Likewise, f2 is represented as p1 (paternal transmitted alleles) throughout the article. A total of 100 randomly picked residual values [$e \sim N(0, I\sigma_e^2)$] were added to genetic values to generate 100 replicates of each simulated phenotype (see methods). * For traits with correlated maternal-fetal genetic effects and POEs, $u_{f1}$ were simulated in the same manner as they were simulated for traits with only fetal effects and POEs.

## Heritability of simulated traits with independent maternal-fetal genetic effects

Traits with independent maternal and fetal effects were simulated in two ways – using the same set and different sets of causal variants in mothers and children. Using independent sets of causal variants and conventional GCTA approach, the estimated $\hat{h}^2$ based on m and f were 24.9% (S.E. = 8.7%) and 27.8% (S.E. = 8.7%) respectively (Fig 2C and S15 Table). Using M-GCTA approach, variance attributable to indirect maternal effect (M'), direct fetal effect (G) and direct-indirect effect covariance (D) were estimated as 22.1% (S.E. = 6.6%), 26.8% (S.E. = 5.6%) and -4.7% (S.E. = 5.5%) respectively (Fig 2C and S15 Table). Conversely, H-GCTA estimated the genetic variance attributable to m1 ($\hat{h}^2_{m1} = 23.4\%$, S.E. = 4.6%), m2 ($\hat{h}^2_{m2} = 12.4\%$, S.E. = 4.8%) and p1 ($\hat{h}^2_{p1} = 11.5\%$, S.E. = 4.2%) (Fig 2C and S15 Table). We observed similar results from traits, simulated using same set of causal variants with independent maternal-fetal genetic effects in mothers and children (Fig 2D and S16 Table). We observed that conventional GCTA and M-GCTA showed equal contribution of maternal and fetal genotypes to the phenotypic variance of the simulated phenotypes. As compared to M-GCTA, H-GCTA estimated the contributions of maternal transmitted (m1), maternal non-transmitted (m2) and paternal transmitted alleles (p1) as expected, i.e., 2:1:1 (S5 Fig and S15 and S16 Tables) which demonstrated equal and independent maternal and fetal contributions.

## Heritability of simulated traits with correlated maternal-fetal genetic effects

We simulated traits influenced by joint maternal-fetal genetic effects with average negative (-0.5, -1.0) and positive (0.5, 1.0) correlation by using same set of causal variants in mothers and children. For traits with 100% negative correlation of maternal-fetal genetic effects, the estimated $\hat{h}^2$ using conventional GCTA approach, were 8.2% (S.E. = 9.4%) and 11.4% (S.E. =

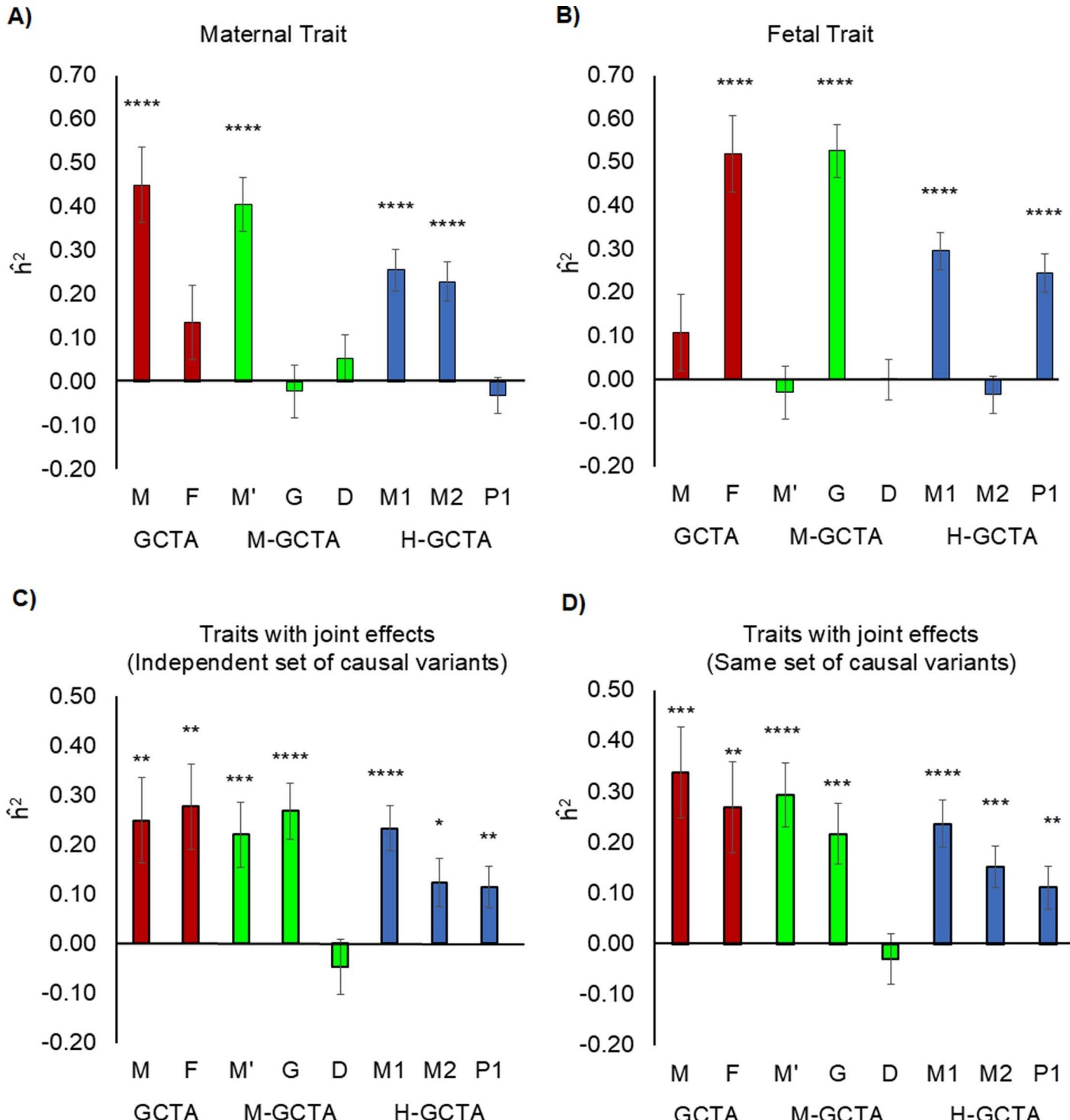

**Fig 2. Comparison of $\hat{h}^2$ for simulated traits from pooled dataset – maternal traits, fetal traits, and traits with independent maternal-fetal genetics effects.** Comparison of $\hat{h}^2$ for simulated traits from pooled dataset, estimated through different approaches fitting GREML (α = -1.0): A) maternal traits; B) fetal traits; C) traits where independent sets of causal variants have effects through mother and fetus; D) traits where same set of causal variants have effects through mother and fetus. For GCTA, M is the GRM generated from maternal genotypes (m), and F is the GRM generated from fetal genotypes (f). For M-GCTA, M' represents the genetic relationship matrix of mothers; G represents genetic relationship matrix of children and D represents mother-child covariance matrix. For H-GCTA, M1 is the GRM generated from maternal transmitted alleles (m1), M2 is the GRM generated from maternal non-transmitted alleles (m2), and P1 is the GRM generated from paternal transmitted alleles (p1). A total of 100 replicates of each phenotype were simulated using empirical genotypes of Pooled dataset. P-values were calculated using z test statistics (two sided). * = (p value <5.0E-02), ** = (p value <1.0E-02), *** = (p value <1.0E-03) and **** = (p value <1.0E-04).

9.4%) based on m and f, respectively (Fig 3A and S17 Table). Using M-GCTA approach, the variance attributable to indirect maternal effect (M'), direct fetal effect (G) and direct-indirect effect covariance (D) were estimated as 35.9% (S.E. = 8.4%), 38.3% (S.E. = 6.8%) and -36.1% (S.E. = 6.2%) respectively (Fig 3A and S17 Table). H-GCTA further partitioned the variance into variance components attributable to m1 ( $\hat{h}^2_{m1} = -1.0\%, \text{S.E.} = 5.0\%$ ), m2 ( $\hat{h}^2_{m2} = 17.6\%, \text{S.E.} = 5.2\%$ ) and p1 ( $\hat{h}^2_{p1} = 20.1\%, \text{S.E.} = 4.1\%$ ) (Fig 3A and S17 Table). Although, negative values of $\check{h}^2$ usually correspond to noise, they are important for interpretation of results for traits with negative correlation of maternal-fetal genetic effects. We observed that conventional GCTA substantially underestimated the genetic contribution of maternal and fetal genomes. As expected, M-GCTA estimated equal contribution of maternal and fetal genetic effects to the phenotypic variance whereas negative and equal contribution of direct-indirect effect covariance suggested 100% negative correlation of maternal-fetal genetic effects. Similarly, H-GCTA showed no contribution from m1 (due to 100% negative correlation of maternal-fetal genetic effects) and almost equal contribution from m2 and p1 to the phenotypic variance (S5 Fig). We observed similar patterns for traits with 50% negative correlation of maternal-fetal genetic effects (Fig 3B and S18 Table).

Similarly, conventional GCTA approach estimated $\hat{h}^2$ based on m and f as 46.2% (S.E. = 8.6%) and 44.8% (S.E. = 8.6%), respectively for traits with 100% positive correlation of maternal-fetal genetic effects (Fig 3C and S19 Table). M-GCTA approach estimated the variance attributable to indirect maternal effect (M'), direct fetal effect (G) and direct-indirect effect covariance (D) as 18.9% (S.E. = 5.0%), 20.6% (S.E. = 5.6%) and 19.7% (S.E. = 4.6%), respectively (Fig 3C and S19 Table). Using H-GCTA, we estimated the genetic variance of simulated traits attributable to m1, m2 and p1 [( $\hat{h}^2_{m1} = 37.4\%, \text{S.E.} = 4.2\%$ ), ( $\hat{h}^2_{m2} = 11.4\%, \text{S.E.} = 3.7\%$ ), ( $\hat{h}^2_{p1} = 11.1\%, \text{S.E.} = 3.9\%$ )] (Fig 3C and S19 Table). While conventional GCTA substantially overestimated the variance attributable to maternal and fetal genotypes, M-GCTA estimated equal contribution of indirect maternal effects, direct fetal effects and direct-indirect effects covariance to the phenotypic variance. Similarly, H-GCTA showed much larger contribution from m1 and equal contribution from m2 and p1 to the phenotypic variance which follows a ratio of 4:1:1 in case of 100% positive correlation of maternal-fetal genetic effects (S5 Fig). Similar patterns were observed for traits with 50% positive correlation of maternal-fetal genetic effects (Fig 3D and S20 Table).

### Heritability of simulated fetal traits with POEs

We also estimated genetic variance using GREML for simulated fetal traits with different levels of parent-of-origin effects (POEs) in varying proportion of causal variants. We simulated two scenarios where maternal imprinting was mimicked by reducing the effect of m1 as compared to p1 in 25% and 50% of the causal variants. In each scenario, we generated a range of imprinting patterns such as $u_{m1}/u_{p1} = 0.75, u_{m1}/u_{p1} = 0.50, u_{m1}/u_{p1} = 0.25 \text{ and } u_{m1}/u_{p1} = 0$. The first three conditions represented partial maternal imprinting whereas the last condition, i.e., $u_{m1}/u_{p1} = 0$ represented complete maternal imprinting. Using our approach (H-GCTA), we estimated the total $\hat{h}^2$ ( $\hat{h}^2_{m1} + \hat{h}^2_{p1}$ ) as expected (~ 50%) (Fig 4 and S21 Table). Results from H-GCTA showed that the variance attributable to m1 ( $\hat{h}^2_{m1}$ ) decreased whereas the variance attributable to p1 ( $\hat{h}^2_{p1}$ ) increased in accordance with the level of imprinting in each scenario (Fig 4A and 4B). We also compared results from our approach with those from GCTA and M-GCTA. While GCTA and M-GCTA were unable to detect contribution of parent-of-origin effects (POEs), H-GCTA detected the variance attributable to POEs as $\hat{h}^2_{m1} - \hat{h}^2_{m2} - \hat{h}^2_{p1}$ (S21 Table).

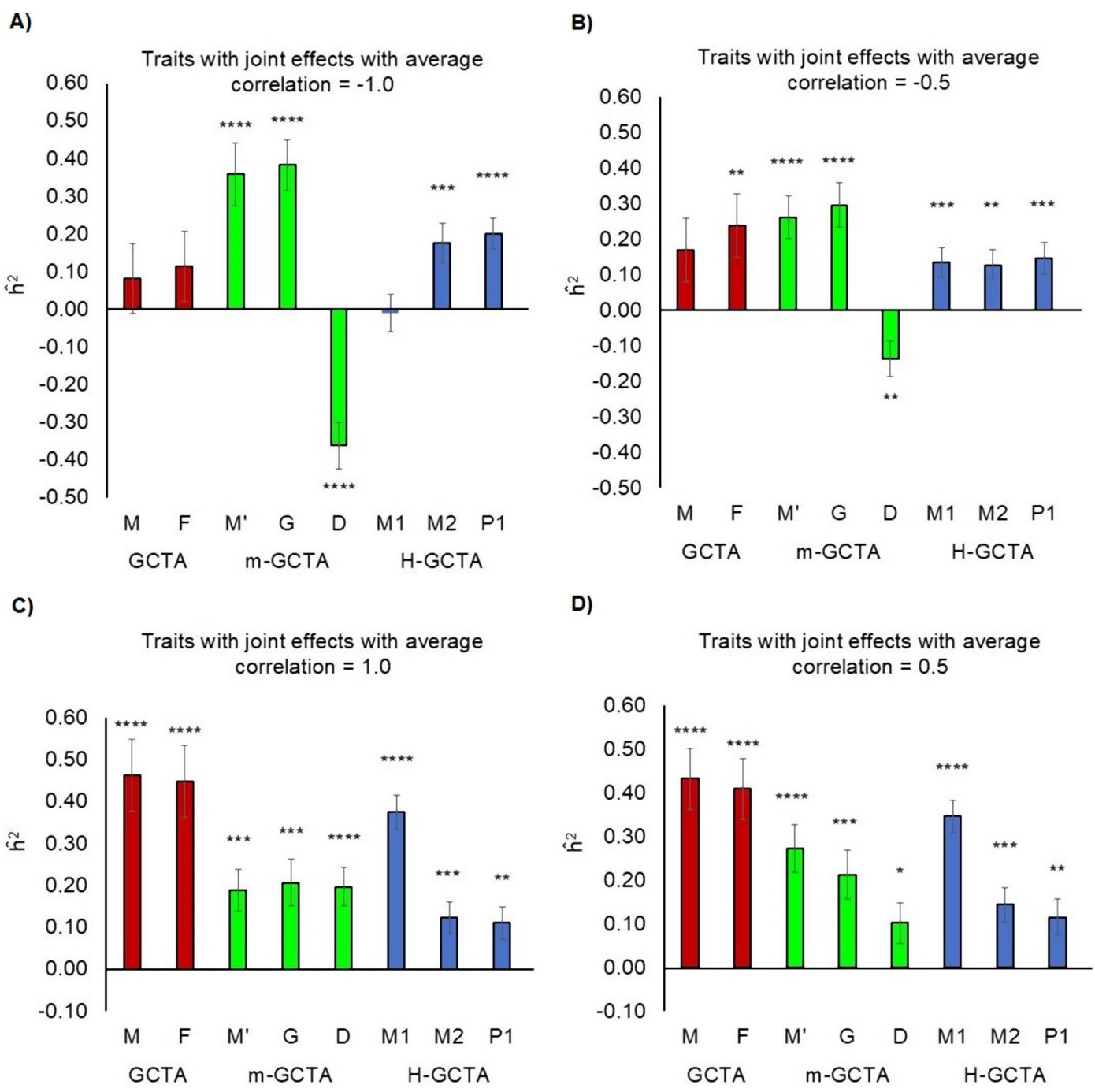

**Fig 3. Comparison of $\hat{h}^2$ for simulated traits from pooled dataset – traits with correlated maternal-fetal genetics effects.** Comparison of $\hat{h}^2$ estimated through different approaches fitting GREML (α = -1.0) for simulated traits with joint maternal–fetal effects from pooled dataset: A) average correlation = -1.0; B) average correlation = -0.5; C) average correlation = 1.0; D) average correlation = 0.5. For GCTA, M is the GRM generated from maternal genotypes (m), and F is the GRM generated from fetal genotypes (f). For M-GCTA, M' represents the genetic relationship matrix of mothers; G represents genetic relationship matrix of children and D represents mother-child covariance matrix. For H-GCTA, M1 is the GRM generated from maternal transmitted alleles (m1), M2 is the GRM generated from maternal non-transmitted alleles (m2), and P1 is the GRM generated from paternal transmitted alleles (p1). A total of 100 replicates of each phenotype were simulated using empirical genotypes of Pooled dataset. P-values were calculated using z test statistics (two sided). * = (p value <5.0E-02), ** = (p value <1.0E-02), *** = (p value <1.0E-03) and **** = (p value <1.0E-04).

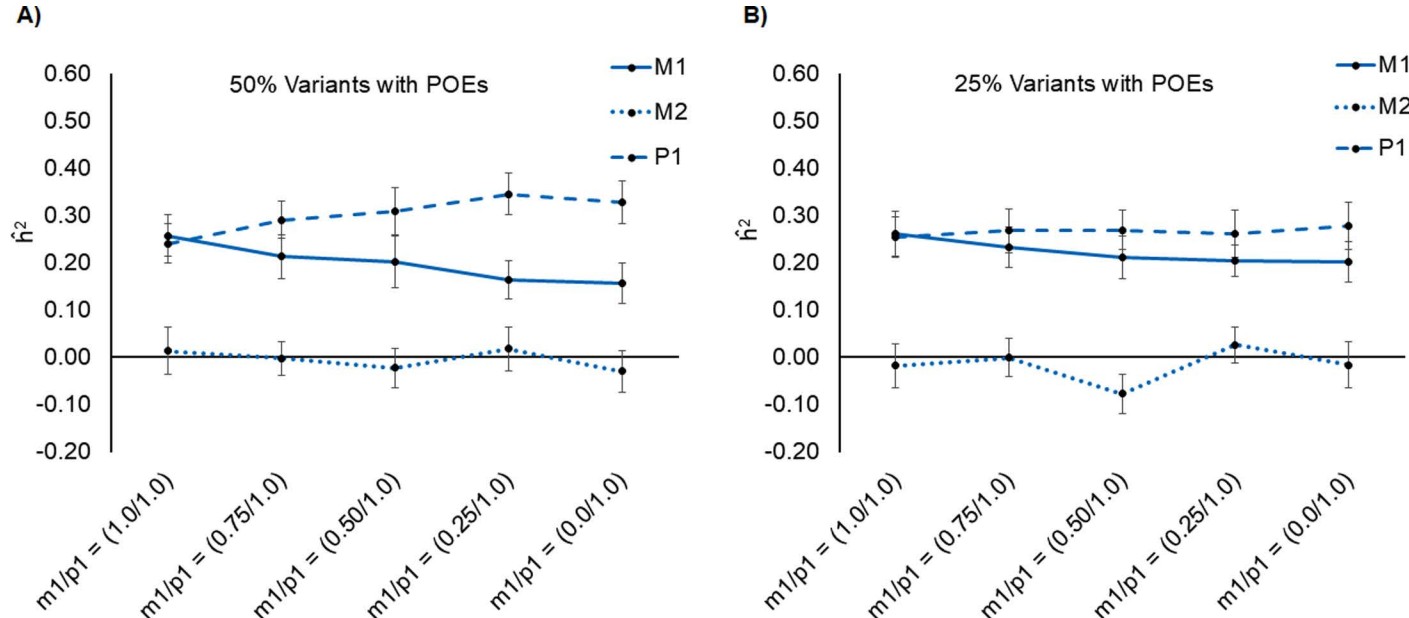

**Fig 4. Comparison of variance attributable to maternal transmitted (m1), maternal non-transmitted (m2), and paternal transmitted (p1) haplotypes for simulated fetal traits with parent-of-origin effects (POEs).** Variance attributable to m1, m2 and p1 estimated through H-GCTA using GREML ($\alpha$ = -1.0) model in simulated fetal traits from pooled dataset: A) 50% causal variants with POEs; B) 25% causal variants with POEs. POEs were incorporated by reducing the effect of m1 as compared to p1 by multiplying effects of m1 with (1 − I) where I is the imprinting factor such as 0.25, 0.50, 0.75 and 1.0. In each scenario, m1 shows either no imprinting, i.e., I = 0.0 ( $m1 / p1 = 1.0 / 1.0$ ) or partial imprinting, i.e., I = 0.25-0.75 ( $m1 / p1 = 0.75 / 1.0 - 0.25 / 1.0$ ) or complete imprinting, i.e., I = 1.0 ( $m1 / p1 = 0.0 / 1.0$ ).

### Heritability of simulated traits with correlated maternal-fetal genetic effects and POEs

To further investigate the intriguing relationships of maternal and fetal genetic effects in dyadic traits, we simulated traits with joint maternal-fetal genetic effects with average correlation = 1.0 and different levels of parent-of-origin effects (POEs) [ $u_{m1} / u_{p1} = 0.75$, $u_{m1} / u_{p1} = 0.50$, $u_{m1} / u_{p1} = 0.25$ and $u_{m1} / u_{p1} = 0$ ]. For simplicity, we assumed that all causal variants exhibited POEs.

For traits with 100% correlated maternal-fetal genetic effects and varying levels of POEs, conventional GCTA approach substantially overestimated $\hat{h}^2$ whereas M-GCTA and H-GCTA approach slightly overestimated $\hat{h}^2$. While GCTA and M-GCTA failed to detect contribution of POEs, our approach (H-GCTA) clearly identified that variance attributable to m1 ( $\hat{h}^2_{m1}$ ) decreases with increasing levels of maternal imprinting and eventually becomes equal to the variance attributable to m2 ( $\hat{h}^2_{m2}$ ) or p1 ( $\hat{h}^2_{p1}$ ) in case of complete maternal imprinting ( $u_{m1} / u_{p1} = 0$ ). As expected, relative variance attributable to m1, m2 and p1 changes from $\hat{h}^2_{m1} : \hat{h}^2_{m2} : \hat{h}^2_{p1} = 4 : 1 : 1$ in the absence of maternal imprinting to $\hat{h}^2_{m1} : \hat{h}^2_{m2} : \hat{h}^2_{p1} = 1 : 1 : 1$ in case of complete maternal imprinting (S5 Fig and S22 Table).

### Heritability estimation of pregnancy-related outcomes using empirical data

All analyses for the estimation of genetic variance were performed using imputed genotype data of ~ 11 million markers across 10,375 mother-child pairs. In addition, three MAF cut-offs (0.001, 0.01 and 0.05) yielding approximately 9 million, 7 million and 5.5 million markers

respectively, were used for analysis. Only independent mother-child pairs (kinship coefficient < 0.05) were used in analysis and 20 principal components (PCs) were used along with genotype-based GRMs in LMM (S6 Fig). For haplotype-based GRMs, we used 30 PCs (10 PCs corresponding to each haplotype) as covariates in LMM (S6 Fig). Like simulated traits, we estimated genetic variance using three approaches – conventional GCTA approach, M-GCTA approach and H-GCTA approach. For each approach, we fitted three models – GREML, LDAK-Thin and LDAK-Weights. Two values of α (-0.25 and -1.0), which represents the extent to which minor allele frequency (MAF) influences the variance of SNP effects on phenotypes were used for each model. Here, we describe results based on GRMs calculated through all polymorphic SNPs and three models with recommended α values, i.e., GREML (α = -1.0), LDAK-Thin (α = -0.25) and LDAK-Weights (α = -0.25). Results based on all polymorphic SNPs using other models are provided in S23 Table. Similarly, results based on GRMs calculated through SNPs with MAF > 0.001, SNPs with MAF > 0.01 and SNPs with MAF > 0.05 are provided in S1 Text and S24, S25 and S26 Tables.

## Heritability of gestational duration

Using GREML (α = -1.0), the conventional GCTA approach estimated $\hat{h}^2$ of gestational duration based on m and f – ($\hat{h}^2_m = 31.4\%$; S.E. = 5.4%) and ($\hat{h}^2_f = 12.2\%$; S.E. = 5.2%). Our approach (H-GCTA) further resolved the variance attributable to m1 – 17.3% (S.E. = 5.2%;), m2 – 12.3% (S.E. = 5.2%) and p1 – 0.0% (S.E. = 5.0%) (Fig 5A and Table 2A). Results using our approach suggested that the genetic variance in gestational duration was primarily influenced by maternal genome, i.e., the SNPs which influence gestational duration through maternal genetic effect. Comparison with M-GCTA confirmed the results from H-GCTA (Fig 5A and Table 2A). The genetic variance estimated through LDAK-Thin (α = -0.25) was similar to those obtained from GREML (α = -1.0). However, estimates from LDAK-Weights (α = -0.25) were substantially larger than those obtained from GREML (α = -1.0) (Table 2A). This pattern is consistent with previous observation for other traits using LDAK model [16] and thoroughly discussed elsewhere [13].

## Heritability of gestational duration adjusted birth weight

Analysis using conventional GCTA showed that the estimated $\hat{h}^2$ of birth weight based on m and f were 16.3% (S.E. = 6.1%) and 34.3% (S.E. = 6.2%) respectively. Using our approach, we further distinguished the variance attributable to m1 – 18.6% (S.E. = 6.1%); m2 – 1.5% (S.E. = 5.6%) and p1 – 13.6% (S.E. = 5.9%) (Fig 5B and Table 2B). The estimates obtained through H-GCTA suggested that genetic variance in birth weight was primarily determined by the fetal genome. Comparison of genetic variance estimated from our approach with those from M-GCTA illustrated that genetic variance in birth weight was mainly attributable to the SNPs which influence birth weight only through direct fetal effect (Fig 5B and Table 2B). Like gestational duration, genetic variance estimated through LDAK-Thin (α = -0.25) was similar to those obtained from GREML (α = -1.0) whereas LDAK-Weights (α = -0.25) estimated larger $\hat{h}^2$.

## Heritability of gestational duration adjusted birth length

We estimated $\hat{h}^2$ of birth length based on m ($\hat{h}^2_m = 23.6\%$; S.E. = 8.3%) and f ($\hat{h}^2_f = 29.0\%$; S.E. = 8.4%) using conventional GCTA approach. While M-GCTA indicated that birth length is largely influenced by positive maternal-fetal covariance ($\hat{h}^2_D = 23.6\%$; S.E. = 8.2%), H-GCTA resolved the genetic variance attributable to m1 – 24.2% (S.E. = 8.4%); m2 – 0.0% (S.E. = 7.9%) and p1 – 4.4% (S.E. = 8.0%) (Fig 5C and Table 2C). H-GCTA showed that unlike birth weight, variance in birth length was mainly attributable to m1 with a much smaller

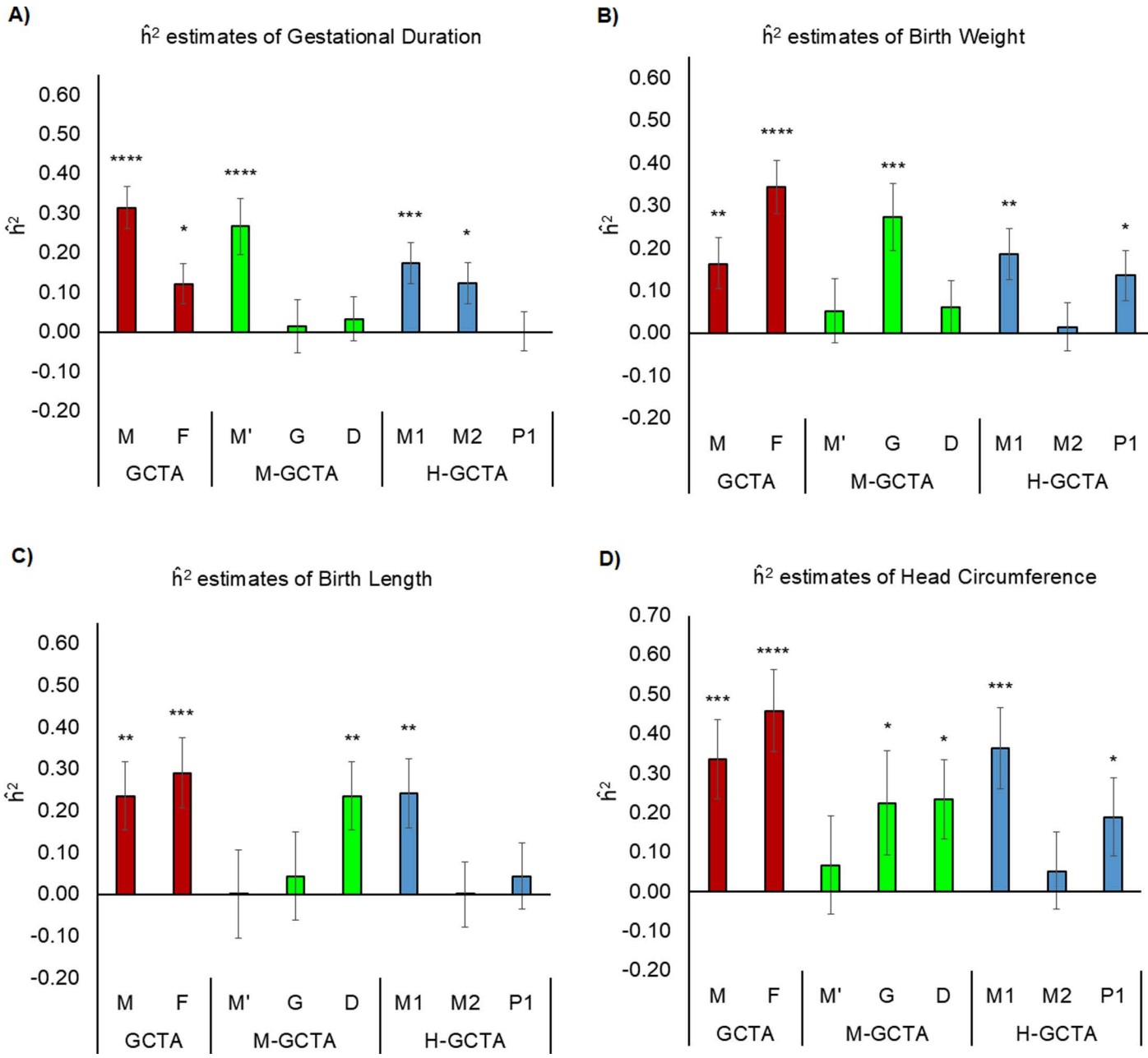

**Fig 5. Comparison of $\hat{h}^2$ for gestational duration and fetal size measurements at birth.** Comparison of $\hat{h}^2$ estimated through different approaches fitting GREML ($\alpha$ = -1.0) for pregnancy-related outcomes in unrelated mother-child pairs (relatedness cutoff > 0.05): A) gestational duration, B) birth weight, C) birth length, and D) head circumference. For GCTA, M is the GRM generated from maternal genotypes (m), and F is the GRM generated from fetal genotypes (f). For M-GCTA, M' represents the genetic relationship matrix of mothers; G represents genetic relationship matrix of children and D represents mother-child covariance matrix. For H-GCTA, M1 is the GRM generated from maternal transmitted alleles (m1), M2 is the GRM generated from maternal non-transmitted alleles (m2), and P1 is the GRM generated from paternal transmitted alleles (p1). For conventional GCTA and M-GCTA approach analyses were adjusted for 20 principal components (PCs) whereas for H-GCTA, analyses were adjusted for 30 PCs (10 PCs corresponding to m1, m2 and p1 each). P-values were calculated using z test statistics (one sided). * = (p value <5.0E-02), ** = (p value <1.0E-02), *** = (p value <1.0E-03) and **** = (p value <1.0E-04).

**Table 2. Comparison of $\hat{h}^2$ for gestational duration and fetal size measurements at birth.**

A) $\hat{h}^2$ of gestational duration

| Approach | GRM | GREML (alpha = -1.0) | | | LDAK-Thin (alpha = -1.0) | | | LDAK-Weights (alpha = -1.0) | | |
|---|---|---|---|---|---|---|---|---|---|---|
| | | $\hat{h}^2$ | SE | p-value | $\hat{h}^2$ | SE | p-value | $\hat{h}^2$ | SE | p-value |
| GCTA | M | 0.3141 | 0.0540 | 3.01E-09 | 0.2787 | 0.0478 | 2.79E-09 | 0.4213 | 0.0901 | 1.46E-06 |
| | F | 0.1217 | 0.0515 | 9.04E-03 | 0.1174 | 0.0467 | 5.96E-03 | 0.2315 | 0.0901 | 5.08E-03 |
| M-GCTA | M' | 0.2670 | 0.0705 | 7.67E-05 | 0.2616 | 0.0628 | 1.56E-05 | 0.3714 | 0.1211 | 1.08E-03 |
| | G | 0.0138 | 0.0673 | 4.19E-01 | 0.0455 | 0.0613 | 2.29E-01 | 0.0835 | 0.1197 | 2.43E-01 |
| | D | 0.0333 | 0.0551 | 2.72E-01 | 0.0097 | 0.0490 | 4.22E-01 | 0.0144 | 0.0984 | 4.42E-01 |
| H-GCTA | M1 | 0.1733 | 0.0523 | 4.64E-04 | 0.1461 | 0.0460 | 7.45E-04 | 0.1107 | 0.0899 | 1.09E-01 |
| | M2 | 0.1226 | 0.0523 | 9.59E-03 | 0.1389 | 0.0468 | 1.49E-03 | 0.1246 | 0.0908 | 8.50E-02 |
| | P1 | 0.0020 | 0.0504 | 4.84E-01 | 0.0534 | 0.0464 | 1.25E-01 | 0.1521 | 0.0904 | 4.62E-02 |

B) $\hat{h}^2$ of birth weight

| Approach | GRM | GREML (alpha = -1.0) | | | LDAK-Thin (alpha = -0.25) | | | LDAK-Weights (alpha = -0.25) | | |
|---|---|---|---|---|---|---|---|---|---|---|
| | | $h^2$ | SE | p-value | $h^2$ | SE | p-value | $h^2$ | SE | p-value |
| GCTA | M | 0.1631 | 0.0605 | 3.52E-03 | 0.1427 | 0.0538 | 3.99E-03 | 0.2908 | 0.1009 | 1.98E-03 |
| | F | 0.3430 | 0.0618 | 1.45E-08 | 0.2700 | 0.0544 | 3.52E-07 | 0.3617 | 0.1020 | 1.96E-04 |
| M-GCTA | M' | 0.0518 | 0.0754 | 2.46E-01 | 0.0694 | 0.0693 | 1.58E-01 | 0.1682 | 0.1348 | 1.06E-01 |
| | G | 0.2721 | 0.0791 | 2.92E-04 | 0.2330 | 0.0710 | 5.15E-04 | 0.2847 | 0.1359 | 1.81E-02 |
| | D | 0.0617 | 0.0608 | 1.55E-01 | 0.0197 | 0.0547 | 3.59E-01 | 0.0244 | 0.1101 | 4.12E-01 |
| H-GCTA | M1 | 0.1861 | 0.0608 | 1.10E-03 | 0.1518 | 0.0539 | 2.42E-03 | 0.2805 | 0.1025 | 3.11E-03 |
| | M2 | 0.0148 | 0.0560 | 3.96E-01 | 0.0195 | 0.0499 | 3.48E-01 | 0.0012 | 0.0992 | 4.95E-01 |
| | P1 | 0.1355 | 0.0588 | 1.06E-02 | 0.1001 | 0.0525 | 2.82E-02 | 0.1087 | 0.1005 | 1.40E-01 |

C) $\hat{h}^2$ of birth length

| Approach | GRM | GREML (alpha = -1.0) | | | LDAK-Thin (alpha = -0.25) | | | LDAK-Weights (alpha = -0.25) | | |
|---|---|---|---|---|---|---|---|---|---|---|
| | | $h^2$ | SE | p-value | $h^2$ | SE | p-value | $h^2$ | SE | p-value |
| GCTA | M | 0.2355 | 0.0829 | 2.26E-03 | 0.1326 | 0.0750 | 3.86E-02 | 0.1905 | 0.1370 | 8.23E-02 |
| | F | 0.2904 | 0.0843 | 2.88E-04 | 0.2318 | 0.0765 | 1.23E-03 | 0.4046 | 0.1398 | 1.90E-03 |
| M-GCTA | M' | 0.0000 | 0.1057 | 5.00E-01 | 0.0093 | 0.0978 | 4.62E-01 | 0.0094 | 0.1861 | 4.80E-01 |
| | G | 0.0426 | 0.1060 | 3.44E-01 | 0.0112 | 0.0970 | 4.54E-01 | 0.0161 | 0.1902 | 4.66E-01 |
| | D | 0.2356 | 0.0823 | 2.11E-03 | 0.1540 | 0.0761 | 2.15E-02 | 0.1982 | 0.1569 | 1.03E-01 |
| H-GCTA | M1 | 0.2415 | 0.0835 | 1.92E-03 | 0.0986 | 0.0737 | 9.05E-02 | 0.2824 | 0.1409 | 2.25E-02 |
| | M2 | 0.0000 | 0.0788 | 5.00E-01 | 0.0087 | 0.0738 | 4.53E-01 | 0.0097 | 0.1393 | 4.72E-01 |
| | P1 | 0.0441 | 0.0802 | 2.91E-01 | 0.0181 | 0.0742 | 4.03E-01 | 0.0131 | 0.1421 | 4.63E-01 |

D) $\hat{h}^2$ of head circumference

| Approach | GRM | GREML (alpha = -1.0) | | | LDAK-Thin (alpha = -0.25) | | | LDAK-Weights (alpha = -0.25) | | |
|---|---|---|---|---|---|---|---|---|---|---|
| | | $h^2$ | SE | p-value | $h^2$ | SE | p-value | $h^2$ | SE | p-value |
| GCTA | M | 0.3351 | 0.1020 | 5.13E-04 | 0.2345 | 0.0907 | 4.85E-03 | 0.4003 | 0.1749 | 1.10E-02 |
| | F | 0.4571 | 0.1049 | 6.61E-06 | 0.3114 | 0.0927 | 3.89E-04 | 0.6998 | 0.1747 | 3.08E-05 |
| M-GCTA | M' | 0.0667 | 0.1268 | 2.99E-01 | 0.0000 | 0.0000 | NA | 0.0070 | 0.2373 | 4.88E-01 |
| | G | 0.2245 | 0.1325 | 4.51E-02 | 0.0915 | 0.1159 | 2.15E-01 | 0.5293 | 0.2339 | 1.18E-02 |
| | D | 0.2343 | 0.1013 | 1.04E-02 | 0.2338 | 0.0729 | 6.69E-04 | 0.1582 | 0.1922 | 2.05E-01 |
| H-GCTA | M1 | 0.3636 | 0.1040 | 2.35E-04 | 0.2029 | 0.0919 | 1.36E-02 | 0.4645 | 0.1738 | 3.76E-03 |
| | M2 | 0.0523 | 0.0985 | 2.98E-01 | 0.0367 | 0.0859 | 3.35E-01 | 0.0467 | 0.1786 | 3.97E-01 |
| | P1 | 0.1879 | 0.1001 | 3.02E-02 | 0.0502 | 0.0896 | 2.88E-01 | 0.1721 | 0.1707 | 1.57E-01 |

*(Continued)*

**Table 2.** (Continued)

Comparison of $\hat{h}^2$ estimated through different approaches fitting GREML (α = -1.0), LDAK-Thin (α = -0.25) and LDAK-Weights (α = -0.25) for A) gestational duration, B) birth weight, C) birth length and D) head circumference. GRMs were generated using all polymorphic SNPs. For GCTA, M is the GRM generated from maternal genotypes (m), and F is the GRM generated from fetal genotypes (f). For M-GCTA, M' represents the genetic relationship matrix of mothers; G represents genetic relationship matrix of children and D represents mother-child covariance matrix. For H-GCTA, M1 is the GRM generated from maternal transmitted alleles (m1), M2 is the GRM generated from maternal non-transmitted alleles (m2), and P1 is the GRM generated from paternal transmitted alleles (p1). Gestational duration was adjusted for fetal sex and fetal size measurements at birth were additionally adjusted for gestational duration up to third orthogonal polynomial. Analyses using GCTA and M-GCTA approach were adjusted for 20 PCs and H-GCTA approach was adjusted for 30 PCs (10 PCs corresponding to m1, m2 and p1 each). P-values were calculated using z test statistics (one sided).

attribution to p1 ($\hat{h}^2_{m1} - \hat{h}^2_{m2} - \hat{h}^2_{p1} \approx 20\%$). According to our simulations, this pattern, i.e., $\hat{h}^2_{m1} > \left(\hat{h}^2_{m2} + \hat{h}^2_{p1}\right)$ could be generated due to either positively correlated maternal-fetal genetic effects (Fig 3 and S19 and S20 Tables) or POEs (Fig 4 and S21 Table). A previous study using M-GCTA [21] suggested that birth length was influenced by both maternal and fetal genome with different genes contributing to the maternal and fetal effects. However, current study indicates that variance in birth length is primarily attributable to positively correlated maternal-fetal genetic effects along with possible POEs [ $(\hat{h}^2_{m1} - \hat{h}^2_{m2}) > \hat{h}^2_{p1}$ ].

## Heritability of gestational duration adjusted head circumference

SNP-based narrow-sense heritability ($\hat{h}^2$) of head circumference estimated using a conventional GCTA approach was 33.5% (S.E. = 10.2%) and 45.7% (S.E. = 10.5%) based on m and f, respectively. Using H-GCTA, we resolved the variance attributable to maternal and fetal genomes into m1 – 36.4% (S.E. = 10.4%); m2 – 5.2% (S.E. = 9.9%) and p1 – 18.8% (S.E. = 10.0%) (Fig 5D and Table 2D). The difference between ($\hat{h}^2_{m1} - \hat{h}^2_{m2}$) and $\hat{h}^2_{p1}$ ($\hat{h}^2_{m1} - \hat{h}^2_{m2} - \hat{h}^2_{p1} \approx 13\%$) suggested that head circumference is largely influenced by fetal genetic effects along with either correlated maternal-fetal genetic effects or possible POEs or both. Similarly, the results from M-GCTA analysis showed approximately equal contribution to variance of head circumference from G and D (Table 2D). The comparison of results from H-GCTA and M-GCTA suggested that head circumference was primarily determined by fetal genome, i.e., phenotypic variance of head circumference was largely influenced by direct fetal effects along with positively correlated joint maternal-fetal effects or POEs. The results also suggested some influence through explicit maternal genetic effect (Table 2D).

## Discussion

Unlike widely studied complex human traits [8,9,13,16,33], pregnancy-related outcomes are simultaneously influenced by maternal and fetal genomes. Therefore, conventional genotype-based approaches that were developed to estimate the genetic contribution to phenotypic variance are limited in addressing the confounding of shared alleles between maternal and fetal genomes. Here, we consider the mother-child pair as a single analytical unit with three haplotypes – maternal transmitted (m1), maternal non-transmitted (m2) and paternal transmitted (p1). Using such an analytical unit, we simultaneously disentangle the contribution of m1 (exclusive and joint maternal-fetal effects), m2 (exclusive maternal effect) and p1 (exclusive fetal effect) to the phenotypic variance. Using the simulated data with varying contributions and correlation of maternal and fetal genetic effects, we show that our newly developed H-GCTA approach can explicitly resolve maternal and fetal contributions and outperforms the GCTA and M-GCTA approach, particularly in the presence of POEs (Fig 4 and S21 Table). We

further apply our haplotype-based approach to distinguish the genetic contribution of mothers and offspring to the phenotypic variance of gestational duration and gestational duration adjusted fetal size measurements at birth in 10,375 European mother-child pairs. A comparison of results from H-GCTA with those from M-GCTA and conventional GCTA approach reveals that gestational duration is primarily influenced by maternal genome whereas fetal size measurements at birth are largely driven by fetal genome. The new results not only confirm the previous findings from epidemiological [34–40] and genetic [21–25,27,31,41–46] studies but also provide new insights into the genetic architecture of fetal size at birth.

The results based on ~11 million polymorphic SNPs show that approximately 17% and 12% variance in gestational duration is attributable to the m1 and m2, respectively with a minimal contribution from p1 (Fig 5 and Table 2). In contrast, variance in gestational duration adjusted fetal size measurements at birth are mainly contributed by m1 ($\hat{h}^2_{m1}$ = 19-36%) and p1 ($\hat{h}^2_{p1}$ = 4-14%) with a minimal contribution from m2 (Fig 5 and Table 2). Among fetal size measurements at birth, variance in birth weight has significant contributions from m1 ($\hat{h}^2_{m1}$ = 19%) as well p1 ($\hat{h}^2_{p1}$ = 14%) whereas variance in birth length and head circumference are mainly attributable to m1 (birth length: $\hat{h}^2_{m1}$ = 24%; head circumference: $\hat{h}^2_{m1}$ = 36%). These new results suggest that variance in gestational duration is mainly attributable to indirect maternal genetic effects whereas variance in birth weight is mainly attributable to direct fetal genetic effects. In addition, a larger contribution of m1 as compared to m2 and p1 ($\hat{h}^2_{m1} - \hat{h}^2_{m2} - \hat{h}^2_{p1} > 0$) to the variance of birth length and head circumference suggests a substantial contribution of correlated maternal-fetal genetic effects or possible POEs or both (Table 2). Results using SNPs with MAF > 0.001, SNPs with MAF > 0.01 and SNPs with MAF > 0.05 showed similar results (S24-S26 Tables).

As observed in the analyses of simulated traits, estimated genetic variance observed through GREML (α = -1.0) and LDAK-Thin (α = -0.25) are similar for pregnancy-related outcomes. Consistent with previous reports [13,16], estimated genetic variance using LDAK-Weights (α = -0.25) are up to 30% higher than those using GREML (α = -1.0). However, analysis using LDAK-Thin (α = -0.25) provides slightly lower estimates for gestational duration and birth weight and substantially lower estimates for birth length and head circumference. Similarly, GREML (α = -0.25) and LDAK-Weights (α = -1.0) estimate substantially smaller $\hat{h}^2$ whereas LDAK-Thin (α = -1.0) estimates substantially larger $\hat{h}^2$ (S23 Table). These results are consistent with the results of simulated traits in the current study and could be due to misspecification of analytical model [47]. For GREML (α = -1.0) model, we observe the largest estimates of genetic variance for each trait using all polymorphic SNPs, which decreases with increasing threshold of MAF cutoff (number of SNPs decrease with increasing MAF cutoff) (S24-S26 Tables). The decrease in the estimated genetic variance with decrease in number of markers is a general limitation of GREML model which is dependent on several assumptions [13,48].

In general, results for pregnancy-related outcomes follow a similar pattern as those for simulated traits. Specifically, estimated genetic variance of gestational duration and birth weight mimic a pattern similar to the simulated maternal and fetal traits, respectively. Interestingly, estimated genetic variance of head circumference follow a mixed pattern with a large fetal and small maternal genetic influence along with a large influence of maternal transmitted alleles. Irrespective of the analytical models and MAF cut-offs for GRM calculation, H-GCTA estimates a larger contribution of m1 as compared to p1 with almost no contribution of m2 to the phenotypic variance of birth length (Tables 2 and S24–S26). Similarly, M-GCTA estimates a larger contribution of correlated maternal-fetal genetic effects (D) as compared to direct fetal effects (G) with almost no contribution of indirect maternal effect (M') (Tables 2 and S24–S26). It is possible that there could be complicated maternal-fetal interactions that are not modeled by any of these approaches.

Interestingly, we observe that the contribution of m1 is larger than m2 or p1 for every pregnancy phenotype in the current study. There are several possible explanations for this pattern of results. The most obvious explanation is that m1 can influence a pregnancy phenotype through both the mother and fetus. For example, for a trait mainly defined by the maternal genome like gestational duration, higher contribution of m1 in comparison to m2 could be due to small but non-zero fetal effect of the m1 alleles. Similarly, for traits mainly defined by the fetal genome such as fetal size measurements at birth, higher contribution of m1 in comparison to p1 could be due to small but non-zero maternal effect of the m1 alleles. Assuming maternal-fetal additivity (genetic effects through mother and fetus influence a pregnancy-related outcome in additive manner) with independent maternal-fetal genetic effects and no POEs, ($\hat{h}^2_{m1}$ - $\hat{h}^2_{m2}$) is equal to $\hat{h}^2_{p1}$. A larger value of $\hat{h}^2_{m1}$ as compared to $\hat{h}^2_{m2}$ and $\hat{h}^2_{p1}$ ($\hat{h}^2_{m1} - \hat{h}^2_{m2} - \hat{h}^2_{p1} > 0$) suggests presence of either positively correlated maternal-fetal genetic effects or possible POEs or both. For birth length and head circumference, the near zero maternal genetic effect (as estimated by M' in the M-GCTA analysis) and the null contribution from the maternal non-transmitted alleles (m2) suggests possible existence of POEs along with positively correlated maternal-fetal genetic effects. Besides the above-mentioned explanations, several other biological phenomena such as interaction between SNPs within the mother or fetus (epistasis) and gene-environment interaction may influence the pattern of genetic variance of pregnancy outcomes.

Despite the above advances, our current approach has certain limitations. Our approach by itself cannot explicitly distinguish the contribution of correlated maternal-fetal genetic effects from POEs. Current haplotype-based approach attempts to relax some of the underlying assumptions in conventional and contemporary approaches such as equal effects of maternal and paternal transmitted alleles in fetus and allelic additivity. However, the interpretation of the results requires assumptions on maternal-fetal additivity and random mating population. In addition, heritability estimation in our approach can also be affected if assumptions such as absence of epistasis (gene-gene interaction) and gene-environment interaction are not met.

In conclusion, we introduce an approach (H-GCTA) to partition phenotypic variance of pregnancy outcomes to maternal transmitted, non-transmitted and paternal transmitted alleles in mother/child pairs. This method provides a direct way to dissect the maternal and fetal genetic contributions to pregnancy-related outcomes. In addition, H-GCTA can be extended to parent-child trios to detect the paternal genetic effect (genetic nurturing effect) [20]. In combination with existing approaches such as M-GCTA and Trio-GCTA [21,24,27], H-GCTA can also be used to resolve the contribution of POEs and correlations between maternal and fetal genetic effects. We believe this approach represents a significant enhance to the genetic analytic toolbox of pregnancy-related outcomes that others will also employ moving forward.

## Methods

### Datasets and quality control

We used genome wide single nucleotide polymorphism (SNP) data from 10,375 mother-child pairs from five European cohorts to distinguish the maternal-fetal genetic contribution to the phenotypic variance of pregnancy-related outcomes such as gestational duration and fetal size measurements at birth (birth weight, birth length and head circumference) (S1 Text and S1 Fig). The study cohorts included Avon Longitudinal Study of Parents and Children (ALSPAC) [49, 50] from UK, Hyperglycemia and Adverse Pregnancy Outcome study (HAPO) [51] from UK, Canada, and Australia, Finnish dataset (FIN) [31,52], Danish Birth Cohort (DNBC) [53], Norwegian Mother, Father and Child Cohort study (MoBa) [54] (S1 Text and S2 Fig and S1-S4 Tables). A detailed description of data sets can be found in supporting data (S1 Text).

Genotyping of DNA extracted from whole blood or swab samples was done on various SNP array platforms such as Affymetrix 6.0, Illumina Human550-Quad, Illumina Human610-Quad, Illumina Human 660W-Quad. SNP array data was filtered based on SNP and sample quality. Quality Control (QC) of genotypes data was performed at two levels – marker level and individual level. Marker level QC was conducted using PLINK 1.9 [55] on the basis of SNP call rate, minor allele frequency (MAF), Hardy-Weinberg Equilibrium (HWE) and individual level QC was done on the basis of call rate per individual, average heterozygosity per individual, sex assignment, inbreeding coefficient. Non-European samples were removed from the study by principal components analysis (PCA) anchored with 1,000 genome samples. Following QC, genotype data of mother-child pairs were phased using SHAPEIT 2 [56]. SHAPEIT 2 automatically recognizes pedigree information provided in the input files. When phasing mother/child duos together, the first allele in child was always the transmitted allele from mother and the second one from father. We imputed the pre-phased genotypes for missing genotypes on Sanger Imputation Server using Positional Burrows-Wheeler Transform (PBWT) software [57]. Haplotype reference consortium (HRC) panel was utilized as reference data for imputation purpose [58]. The phasing and mother-child allele transmission of the imputed alleles were retained from the pre-phasing stage.

QC of phenotype data was conducted considering gestational duration as the primary outcome. Pregnancies involving history of risk factors for preterm birth or any medical complication during pregnancy influencing preterm birth, C-sections and non-spontaneous births were excluded. We also excluded, non-singlet pregnancies, pregnancies who self-reported non-European ancestry and children who could not survive > 1 year. Additionally, gestational duration was adjusted for fetal sex; fetal size measurements at birth such as birth weight, birth length and head circumference were adjusted for gestational duration up to third orthogonal polynomial component. Details of genotype and phenotype QC is provided in supporting data (S1 Text).

## Statistical method

We used a linear mixed model (LMM) to estimate the SNP-heritability ($\hat{h}^2$) of simulated and empirical phenotypes. This model assumes that the phenotype was normally distributed - Y ~ N(μ, V) with mean μ and variance V. We created GRMs from standardized genotypes/haplotypes utilizing the method developed by Yang et.al. [7, 8] and Speed et. al. [9,16]. Each cell of the genotype-based GRM and haplotype- based GRM represented relatedness between two individuals j and k calculated based on genotypes (Equation 1) and haplotypes (Equation 2) respectively.

$$A_{jk} = \frac{1}{S}\sum_{i=1}^{s}\frac{\left(x_{ij} - 2p_i\right)\left(x_{ik} - 2p_i\right)}{2p_i\left(1 - p_i\right)} \tag{1}$$

Where, $A_{jk}$ is the correlation coefficient between two individuals j and k averaged over all SNPs; S is number of SNPs used to calculate relatedness; $x_{ij}$ is the number of copies of the reference alleles in individual j for SNP i (i.e., 0 or 1 or 2); $x_{ik}$ is the number of copies of the reference alleles in individual k for SNP i (0 or 1 or 2); $p_i$ is frequency of reference allele of SNP i.

$$T_{jk} = \frac{1}{S}\sum_{i=1}^{s}\frac{\left(c_{ij} - p_i\right)\left(c_{ik} - p_i\right)}{p_i\left(1 - p_i\right)} \tag{2}$$

Where, $T_{jk}$ is the correlation coefficient between two mother/child duos or full trios j and k based on maternal transmitted alleles (m1) or maternal non-transmitted alleles (m2) or

paternal transmitted alleles (p1) or paternal non-transmitted alleles (p2); S is number of SNPs whose alleles are used to calculate relatedness; $c_{ij}$ is the number of the reference alleles of m1 or m2 or p1 or p2 in mother/child duo or full trio j for SNP i (i.e., 0 or 1); $c_{ik}$ is the number of the reference alleles of m1 or m2 or p1 or p2 in mother/child duo or full trio k for SNP i (i.e., 0 or 1); $p_i$ is frequency of reference allele of SNP i.

For genotype-based analysis, we created two GRMs - M and F by utilizing maternal genotypes (m) and fetal genotypes (f) respectively. For haplotype-based analysis, we considered mother-child pair as a single analytical unit consisting of three haplotypes corresponding to m1, m2, and p1. We created three separate GRMs - M1, M2 and P1 using only m1, only m2 and only p1 respectively (Fig 1A and 1B). We fitted mothers' genotype-based GRM (M) (Equations 3 and 4) and children's genotype-based GRM (F) (Equations 5 and 6) separately in LMM to estimate phenotypic variance attributable to maternal and fetal genotypes respectively. To calculate explicit contribution of maternal and fetal genomes to the overall narrow-sense heritability, we simultaneously fitted all three matrices (M1, M2 and P1) in LMM and estimated the additive genetic variance attributable to each of the three components (Equation 7, 8).

$$Y_s = X\beta + Z_m u_m + e \tag{3}$$

$$Y_s Y_s^{'} = XX^{'}\sigma_{\dagger}^2 + M\sigma_M^2 + I\sigma_e^2 \tag{4}$$

$$Y_s = X\beta + Z_f u_f + e \tag{5}$$

$$Y_s Y_s^{'} = XX^{'}\sigma_{\beta}^2 + F\sigma_F^2 + I\sigma_e^2 \tag{6}$$

$$Y_s = X\beta + Z_{m1} u_{m1} + Z_{m2} u_{m2} + Z_{p1} u_{p1} + e \tag{7}$$

$$Y_s Y_s^{'} = XX^{'}\sigma_{\dagger}^2 + M1\sigma_{M1}^2 + M2\sigma_{M2}^2 + P1\sigma_{P1}^2 + I\sigma_e^2 \tag{8}$$

Where, $Y_s$ is a vector of standardized phenotype (n x 1; where, n is number of individuals); X is a matrix of covariates representing fixed effects (n x p; where, p is number of fixed effects); β is a vector of fixed effects (p x 1); $Z_m$ is a matrix of mothers' standardized genotypes (m) (n x S; where, S is number of SNPs); $Z_f$ is a matrix of children's standardized genotypes (f) (n x S); $Z_{m1}$ is a matrix of standardized maternal transmitted alleles (m1) (n x S); $Z_{m2}$ is a matrix of standardized maternal non-transmitted alleles (m2) (n x S); $Z_{p1}$ is a matrix of standardized paternal transmitted alleles (p1) (n x S); ε is a vector of residual effects with e ~ N(0, $I\sigma_e^2$); $u_m$ and $u_f$ are vectors of random effect sizes for maternal genotypes (m) and fetal genotypes (f); $u_{m1}$, $u_{m2}$ and $u_{p1}$ are vectors of random effect sizes for maternal transmitted (m1), maternal non-transmitted (m2) and paternal transmitted (p1) alleles respectively (m x 1); $Y_s Y_s^{'}$ is Variance-Covariance matrix of phenotypes; M, F, M1, M2 and P1 are GRMs generated from $Z_m$, $Z_f$, $Z_{m1}$, $Z_{m2}$ and $Z_{p1}$ respectively (e.g., $M1 = \frac{1}{S} * Z_{m1} Z_{m1}^{'}$); $\sigma^2$ are the variances of the respective components.

As previously reported(9, 16, 47), genetic architecture is parametrized on MAF and pair-wise LD, assuming $E[\text{var}(u_i)] \sim w_i [p_i(1 - p_i)]^{1+\alpha}$, where $u_i$, $w_i$ and $p_i$ are the effect size, weight and reference allele frequency of SNP i and α is the scaling factor which represents the extent to which MAF influences the variance of per-allele effect of SNP i [var($u_i$)]. We calculated SNP-specific weights using LDAK and scaled GRMs with two α values ($\alpha = -0.25$ and $-1.0$) in each model. Each standardized column of genotype/haplotype matrix (n x S) was multiplied by $w_i[p_i(1 - p_i)]^{1+\alpha}$ ($\alpha = -0.25, -1.0$) before fitting into LMM.

## Implementation

Phenotypic variance, i.e., Var(Y) attributable to different components could be estimated by fitting GRMs corresponding to those components in LMM. We used REML implemented through GCTA(7, 8) and LDAK [9,16] to estimate $\hat{h}^2$ of simulated and empirical phenotypes. For genotype-based analysis through conventional GCTA approach [7], we fitted a GRM generated from mothers' genotypes (M) and children's genotypes (F) separately in LMM whereas for haplotype-based analysis through H-GCTA approach, we fitted three GRMs (M1, M2 and P1) simultaneously in LMM. We also compared results from our approach with those from a contemporary approach, M-GCTA [21,24]. Analysis through the M-GCTA approach involved generation of the GRMs using mothers' and children's genotypes together. The upper left quadrant of the GRM represented genetic relationship matrix of mothers (M'); the lower right quadrant represented genetic relationship matrix of children (G) and sum of the lower left quadrant and its transpose represented the genetic relationship matrix of mothers and children (D).

Each approach was fitted through three different models, namely, GREML, LDAK-Thin (where, all pruned SNPs with $r^2 \leq 0.98$ were given equal weights, i.e., 1.0) and LDAK-Weights (where, specific weights were calculated for each pruned SNP based on its pair-wise LD with other SNPs in a 100 kb window) (S1 Fig). For GREML and LDAK-Thin model, constant values of $w_i$ were used ($w_i = 1$). The difference between GREML and LDAK-Thin model exists in the number of SNPs used to calculate GRM. While GREML uses all genotyped/imputed SNPs to calculate GRM, LDAK-Thin uses only pruned SNPs for the same. On the other hand, LDAK-Weights model uses SNP-specific weights along with specific values of $\alpha$ for scaling.

## Simulation

A total of 100 replicates of phenotypes were simulated using empirical genotype data from 10,375 mother-child pairs. We randomly selected 10,000 causal variants from a common set of all polymorphic SNPs across all datasets (approximately 11 million markers) and randomly picked their effect sizes from standard normal distribution [N(0,1)]. Phenotypes were generated from the model $y_j = g_j + e_j$, where $y_j$, $g_j$ and $e_j$ are phenotypic, genetic and residual (environmental) values for individual j. Genetic value of individual j was calculated as $g_j = \sum_i Z_{ij} u_i$ where $Z_{ij}$ is standardized genotypic value and $u_i$ is effect size of variant i in individual j. Multiplication of randomly picked effect sizes [($u_i \sim (0,1)$] with standardized genotype/haplotype matrix implies that effect sizes are inversely proportional to MAF. A total of 100 independently generated residual values were added to individual's genetic value ($g_j$) to simulate 100 replicates of phenotype. Residual effects were randomly drawn from a distribution $e \sim N(0, I\sigma_e^2)$ where e is a vector of residual effects, I is an identity matrix and $\sigma_e^2$ is the variance of residual effects with $\sigma_e^2 = \sigma_g^2 \left( \frac{1}{\hat{h}^2} - 1 \right)$ where $\sigma_g^2$ is the variance of genetic values and $\hat{h}^2$ is a preset SNP-based narrow-sense heritability ($\hat{h}^2 = 0.5$).

Three types of traits were simulated considering effects only from the mother (maternal traits), only from fetus (fetal traits) and joint maternal-fetal effects (Table 1). Traits with joint maternal-fetal effects were simulated with different levels of average correlation among maternal and fetal genetic effects (-0.5, -1.0, 0.5 and 1.0) (Table 1). First, traits with independent maternal-fetal genetic effects were simulated using independent and same set sets of causal variants in mothers and children. As we observed similar results in both scenarios, traits with correlated maternal-fetal genetic effects were simulated using same set of 10,000 causal variants in mothers and children. We also simulated traits with POEs, where m1 had less effect in comparison to p1. We considered different scenarios, where varying fractions of causal variants, e.g., 25%, 50%, showed maternal imprinting. In each scenario, we simulated

different levels of imprinting for m1 (25% - 100%) by reducing effect sizes of m1 (75% - 0%) as compared to p1 (Table 1). Non-zero effects of m1 as compared to p1 represented partial maternal imprinting whereas no effect of m1 represented complete imprinting. All relatedness matrices using simulated data were generated and fitted using different models such as GREML, LDAK-Thin and LDAK-weights into LMM in a similar way as mentioned in the statistical method and Implementation section. We compared our haplotype-based approach (H-GCTA) with conventional GCTA approach and a contemporary M-GCTA approach using above mentioned models with two α values (-1.0, -0.25) for all simulated traits. All analyses for simulated data were run using unrestricted REML, i.e., $\hat{h}^2$ estimates could be less than zero.

## Analysis of empirical datasets

We performed analyses using three sets of markers – all polymorphic SNPs, SNPs with MAF > 0.001, SNPs with MAF > 0.01 and SNPs with MAF > 0.05, to include the contribution of very rare, rare, common and very common variants to the heritability of pregnancy-related outcomes (S1 Fig). The marker sets based on the MAF cutoff were selected in each dataset separately, considering mothers as founders. Then, a common set of markers across all datasets was selected in each MAF cutoff category. We pooled individual datasets and generated five different GRMs utilizing mothers' genotypes (M), children's genotypes (F), maternal transmitted haplotypes (M1), maternal non-transmitted haplotypes (M2) and paternal transmitted haplotypes (P1) using the imputed genotype data of mother/child pairs (S2 Table). One of the related individuals was removed from each GRM (relatedness coefficient > 0.05) and a common set of mother-child pairs across five GRMs was selected in each MAF cutoff category (S3 Table). The GRMs were created and fitted into LMM using GREML, LDAK-Thin and LDAK-Weights model. To avoid the problem of "non positive definite variance-covariance matrix" and "non-convergence of likelihood" particularly in models with multiple GRMs, LMM-based analyses for empirical data were performed using restricted REML, i.e., $\hat{h}^2$ estimates could not be less than zero, except for the analyses with very common SNPs (MAF > 0.05). All the analyses were adjusted for principal components (PCs) – 20 PCs for analyses through GCTA and M-GCTA and 30 PCs (10 PCs corresponding to m1, m2 and p1 each) for analyses through H-GCTA (S6 Fig). We also replicated our findings in another Nordic dataset (HARVEST) of ~ 8,000 mother-child pairs (S1 Text). We estimated the $\hat{h}^2$ of gestational duration through GREML (α = -1.0) in replication dataset using SNPs with MAF > 0.01 (S7 Fig and S27 Table).

## Supporting information

**S1 Text.  Supporting data.**
(DOCX)

**S1 Table.  Genotype and Phenotype records in datasets.** Number of pregnancies and genotypes present in individual datasets. a) Number of genotypes typed in each dataset b) number of genotypes passed through genotype QC; c) number of pregnancies after genotype QC and phenotype inclusion/exclusion.
(DOCX)

**S2 Table.  Genotype information in individual datasets.** Number of imputed sites using Haplotype Reference Consortium (HRC), polymorphic SNPs in individual datasets and common set of SNPs across all available datasets. In individual datasets, mothers were considered as founders for each MAF cutoff category and corresponding children were selected. Final analysis was performed using pooled data and common set of SNPs across all datasets.
(DOCX)

**S3 Table. Phenotype information in pooled dataset.** Number of samples with gestational duration, birth weight, birth length and head circumference in the pooled data; a) without relatedness coefficient cut-off, b) with relatedness coefficient cut-off < 0.05. For mother-child pairs with relatedness coefficient cutoff < 0.05, common set of mother-child pairs were selected from GRMs based on mother's genotypes, children's genotypes, maternal transmitted alleles (m1), maternal non-transmitted alleles (m2) and paternal transmitted alleles (p1). (DOCX)

**S4 Table. Phenotype summary in individual datasets.** Descriptive statistics of gestational duration, birth weight, birth length and head circumference in ALSPAC, HAPO, FIN, DNBC and MoBa. All four traits were available only in two datasets, namely ALSPAC and HAPO. (DOCX)

**S5 Table. SNP-based heritability of simulated maternal traits from ALSPAC dataset.** $\hat{h}^2$ of simulated maternal traits from ALSPAC dataset, estimated through conventional GCTA, M-GCTA and H-GCTA approach. Each approach was fitted using GREML ($\alpha$ = -0.25, -1.0), LDAK-Thin ($\alpha$ = -0.25, -1.0) and LDAK-Weights ($\alpha$ = -0.25, -1.0). For GCTA, M is the GRM generated from maternal genotypes (m), and F is the GRM generated from fetal genotypes (f). For M-GCTA, M' represents the genetic relationship matrix of mothers; G represents genetic relationship matrix of children and D represents mother-child covariance matrix. For H-GCTA, M1 is the GRM generated from maternal transmitted alleles (m1), M2 is the GRM generated from maternal non-transmitted alleles (m2), and P1 is the GRM generated from paternal transmitted alleles (p1). A total of 100 replicates of each phenotype were simulated using empirical genotypes of ALSPAC dataset. P-values were calculated using z test statistics (two sided).
(DOCX)

**S6 Table. SNP-based heritability of simulated fetal traits from ALSPAC dataset.** $\hat{h}^2$ of simulated fetal traits from ALSPAC dataset, estimated through conventional GCTA, M-GCTA and H-GCTA approach. Each approach was fitted using GREML ($\alpha$ = -0.25, -1.0), LDAK-Thin ($\alpha$ = -0.25, -1.0) and LDAK-Weights ($\alpha$ = -0.25, -1.0). For GCTA, M is the GRM generated from maternal genotypes (m), and F is the GRM generated from fetal genotypes (f). For M-GCTA, M' represents the genetic relationship matrix of mothers; G represents genetic relationship matrix of children and D represents mother-child covariance matrix. For H-GCTA, M1 is the GRM generated from maternal transmitted alleles (m1), M2 is the GRM generated from maternal non-transmitted alleles (m2), and P1 is the GRM generated from paternal transmitted alleles (p1). A total of 100 replicates of each phenotype were simulated using empirical genotypes of ALSPAC dataset. P-values were calculated using z test statistics (two sided).
(DOCX)

**S7 Table. SNP-based heritability of simulated traits from ALSPAC dataset with independent maternal-fetal genetic effects using independent sets of causal variants in mother and child.** $\hat{h}^2$ of simulated traits from ALSPAC dataset with independent maternal-fetal genetic effects (independent sets of causal variants in mother and child), estimated through conventional GCTA, M-GCTA and H-GCTA approach. Each approach was fitted using GREML ($\alpha$ = -0.25, -1.0), LDAK-Thin ($\alpha$ = -0.25, -1.0) and LDAK-Weights ($\alpha$ = -0.25, -1.0). For GCTA, M is the GRM generated from maternal genotypes (m), and F is the GRM generated from fetal genotypes (f). For M-GCTA, M' represents the genetic relationship matrix of mothers; G represents genetic relationship matrix of children and D represents mother-child covariance matrix. For H-GCTA, M1 is the GRM generated from maternal transmitted alleles (m1), M2

is the GRM generated from maternal non-transmitted alleles (m2), and P1 is the GRM generated from paternal transmitted alleles (p1). A total of 100 replicates of each phenotype were simulated using empirical genotypes of ALSPAC dataset. P-values were calculated using z test statistics (two sided).
(DOCX)

**S8 Table. SNP-based heritability of simulated traits from ALSPAC dataset with independent maternal-fetal genetic effects using same set of causal variants in mother and child.** $\hat{h}^2$ of simulated traits from ALSPAC dataset with independent maternal-fetal genetic effects (same set of causal variants in mother and child), estimated through conventional GCTA, M-GCTA and H-GCTA approach. Each approach was fitted using GREML (α = -0.25, -1.0), LDAK-Thin (α = -0.25, -1.0) and LDAK-Weights (α = -0.25, -1.0). For GCTA, M is the GRM generated from maternal genotypes (m), and F is the GRM generated from fetal genotypes (f). For M-GCTA, M' represents the genetic relationship matrix of mothers; G represents genetic relationship matrix of children and D represents mother-child covariance matrix. For H-GCTA, M1 is the GRM generated from maternal transmitted alleles (m1), M2 is the GRM generated from maternal non-transmitted alleles (m2), and P1 is the GRM generated from paternal transmitted alleles (p1). A total of 100 replicates of each phenotype were simulated using empirical genotypes of ALSPAC dataset. P-values were calculated using z test statistics (two sided).
(DOCX)

**S9 Table. SNP-based heritability of simulated traits from ALSPAC dataset with correlated maternal-fetal genetic effects (average correlation = -1.0).** $\hat{h}^2$ of simulated traits with correlated maternal-fetal genetic effects (average correlation = -1.0), estimated through conventional GCTA, M-GCTA and H-GCTA approach. Each approach was fitted using GREML (α = -0.25, -1.0), LDAK-Thin (α = -0.25, -1.0) and LDAK-Weights (α = -0.25, -1.0). For GCTA, M is the GRM generated from maternal genotypes (m), and F is the GRM generated from fetal genotypes (f). For M-GCTA, M' represents the genetic relationship matrix of mothers; G represents genetic relationship matrix of children and D represents mother-child covariance matrix. For H-GCTA, M1 is the GRM generated from maternal transmitted alleles (m1), M2 is the GRM generated from maternal non-transmitted alleles (m2), and P1 is the GRM generated from paternal transmitted alleles (p1). A total of 100 replicates of each phenotype were simulated using empirical genotypes of ALSPAC dataset. P-values were calculated using z test statistics (two sided).
(DOCX)

**S10 Table. SNP-based heritability of simulated traits from ALSPAC dataset with correlated maternal-fetal genetic effects (average correlation = -0.5).** $\hat{h}^2$ of simulated traits from ALSPAC dataset with correlated maternal-fetal genetic effects (average correlation = -0.5), estimated through conventional GCTA, M-GCTA and H-GCTA approach. Each approach was fitted using GREML (α = -0.25, -1.0), LDAK-Thin (α = -0.25, -1.0) and LDAK-Weights (α = -0.25, -1.0). For GCTA, M is the GRM generated from maternal genotypes (m), and F is the GRM generated from fetal genotypes (f). For M-GCTA, M' represents the genetic relationship matrix of mothers; G represents genetic relationship matrix of children and D represents mother-child covariance matrix. For H-GCTA, M1 is the GRM generated from maternal transmitted alleles (m1), M2 is the GRM generated from maternal non-transmitted alleles (m2), and P1 is the GRM generated from paternal transmitted alleles (p1). A total of 100 replicates of each phenotype were simulated using empirical genotypes of ALSPAC dataset. P-values were calculated using z test statistics (two sided).
(DOCX)

**S11 Table. SNP-based heritability of simulated traits from ALSPAC dataset with correlated maternal-fetal genetic effects (average correlation = 1.0).** $\hat{h}^2$ of simulated traits from ALSPAC dataset with correlated maternal-fetal genetic effects (average correlation = 1.0), estimated through conventional GCTA, M-GCTA and H-GCTA approach. Each approach was fitted using GREML (α = -0.25, -1.0), LDAK-Thin (α = -0.25, -1.0) and LDAK-Weights (α = -0.25, -1.0). For GCTA, M is the GRM generated from maternal genotypes (m), and F is the GRM generated from fetal genotypes (f). For M-GCTA, M' represents the genetic relationship matrix of mothers; G represents genetic relationship matrix of children and D represents mother-child covariance matrix. For H-GCTA, M1 is the GRM generated from maternal transmitted alleles (m1), M2 is the GRM generated from maternal non-transmitted alleles (m2), and P1 is the GRM generated from paternal transmitted alleles (p1). A total of 100 replicates of each phenotype were simulated using empirical genotypes of ALSPAC dataset. P-values were calculated using z test statistics (two sided).
(DOCX)

**S12 Table. SNP-based heritability of simulated traits from ALSPAC dataset with correlated maternal-fetal genetic effects (average correlation = 0.5).** $\hat{h}^2$ of simulated traits from ALSPAC dataset with correlated maternal-fetal genetic effects (average correlation = 0.5), estimated through conventional GCTA, M-GCTA and H-GCTA approach. Each approach was fitted using GREML (α = -0.25, -1.0), LDAK-Thin (α = -0.25, -1.0) and LDAK-Weights (α = -0.25, -1.0). For GCTA, M is the GRM generated from maternal genotypes (m), and F is the GRM generated from fetal genotypes (f). For M-GCTA, M' represents the genetic relationship matrix of mothers; G represents genetic relationship matrix of children and D represents mother-child covariance matrix. For H-GCTA, M1 is the GRM generated from maternal transmitted alleles (m1), M2 is the GRM generated from maternal non-transmitted alleles (m2), and P1 is the GRM generated from paternal transmitted alleles (p1). A total of 100 replicates of each phenotype were simulated using empirical genotypes of ALSPAC dataset. P-values were calculated using z test statistics (two sided).
(DOCX)

**S13 Table. SNP-based heritability of simulated maternal traits from pooled dataset.** $\hat{h}^2$ of simulated maternal traits from pooled dataset, estimated through conventional GCTA, M-GCTA and H-GCTA approach. Each approach was fitted using GREML (α = -0.25, -1.0), LDAK-Thin (α = -0.25, -1.0) and LDAK-Weights (α = -0.25, -1.0). For GCTA, M is the GRM generated from maternal genotypes (m), and F is the GRM generated from fetal genotypes (f). For M-GCTA, M' represents the genetic relationship matrix of mothers; G represents genetic relationship matrix of children and D represents mother-child covariance matrix. For H-GCTA, M1 is the GRM generated from maternal transmitted alleles (m1), M2 is the GRM generated from maternal non-transmitted alleles (m2), and P1 is the GRM generated from paternal transmitted alleles (p1). A total of 100 replicates of each phenotype were simulated using empirical genotypes of Pooled dataset. P-values were calculated using z test statistics (two sided).
(DOCX)

**S14 Table. SNP-based heritability of simulated fetal traits from pooled dataset.** $\hat{h}^2$ of simulated fetal traits from pooled dataset, estimated through conventional GCTA, M-GCTA and H-GCTA approach. Each approach was fitted using GREML (α = -0.25, -1.0), LDAK-Thin (α = -0.25, -1.0) and LDAK-Weights (α = -0.25, -1.0). For GCTA, M is the GRM generated from maternal genotypes (m), and F is the GRM generated from fetal genotypes (f). For M-GCTA, M' represents the genetic relationship matrix of mothers; G represents genetic relationship

matrix of children and D represents mother-child covariance matrix. For H-GCTA, M1 is the GRM generated from maternal transmitted alleles (m1), M2 is the GRM generated from maternal non-transmitted alleles (m2), and P1 is the GRM generated from paternal transmitted alleles (p1). A total of 100 replicates of each phenotype were simulated using empirical genotypes of Pooled dataset. P-values were calculated using z test statistics (two sided).
(DOCX)

**S15 Table. SNP-based heritability of simulated traits from pooled dataset with independent maternal-fetal genetic effects using independent sets of causal variants in mother and child.** $\hat{h}^2$ of simulated traits from pooled dataset with independent maternal-fetal genetic effects (independent sets of causal variants in mother and child), estimated through conventional GCTA, M-GCTA and H-GCTA approach. Each approach was fitted using GREML ($\alpha$ = -0.25, -1.0), LDAK-Thin ($\alpha$ = -0.25, -1.0) and LDAK-Weights ($\alpha$ = -0.25, -1.0). For GCTA, M is the GRM generated from maternal genotypes (m), and F is the GRM generated from fetal genotypes (f). For M-GCTA, M' represents the genetic relationship matrix of mothers; G represents genetic relationship matrix of children and D represents mother-child covariance matrix. For H-GCTA, M1 is the GRM generated from maternal transmitted alleles (m1), M2 is the GRM generated from maternal non-transmitted alleles (m2), and P1 is the GRM generated from paternal transmitted alleles (p1). A total of 100 replicates of each phenotype were simulated using empirical genotypes of Pooled dataset. P-values were calculated using z test statistics (two sided).
(DOCX)

**S16 Table. SNP-based heritability of simulated traits from pooled dataset with independent maternal-fetal genetic effects using same set of causal variants in mother and child.** $\hat{h}^2$ of simulated traits from pooled dataset with independent maternal-fetal genetic effects (same set of causal variants in mother and child), estimated through conventional GCTA, M-GCTA and H-GCTA approach. Each approach was fitted using GREML ($\alpha$ = -0.25, -1.0), LDAK-Thin ($\alpha$ = -0.25, -1.0) and LDAK-Weights ($\alpha$ = -0.25, -1.0). For GCTA, M is the GRM generated from maternal genotypes (m), and F is the GRM generated from fetal genotypes (f). For M-GCTA, M' represents the genetic relationship matrix of mothers; G represents genetic relationship matrix of children and D represents mother-child covariance matrix. For H-GCTA, M1 is the GRM generated from maternal transmitted alleles (m1), M2 is the GRM generated from maternal non-transmitted alleles (m2), and P1 is the GRM generated from paternal transmitted alleles (p1). A total of 100 replicates of each phenotype were simulated using empirical genotypes of Pooled dataset. P-values were calculated using z test statistics (two sided).
(DOCX)

**S17 Table. SNP-based heritability of simulated traits from pooled dataset with correlated maternal-fetal genetic effects (average correlation = -1.0).** $\hat{h}^2$ of simulated traits from pooled dataset with correlated maternal-fetal genetic effects (average correlation = -1.0), estimated through conventional GCTA, M-GCTA and H-GCTA approach. Each approach was fitted using GREML ($\alpha$ = -0.25, -1.0), LDAK-Thin ($\alpha$ = -0.25, -1.0) and LDAK-Weights ($\alpha$ = -0.25, -1.0). For GCTA, M is the GRM generated from maternal genotypes (m), and F is the GRM generated from fetal genotypes (f). For M-GCTA, M' represents the genetic relationship matrix of mothers; G represents genetic relationship matrix of children and D represents mother-child covariance matrix. For H-GCTA, M1 is the GRM generated from maternal transmitted alleles (m1), M2 is the GRM generated from maternal non-transmitted alleles (m2), and P1 is the GRM generated from paternal transmitted alleles (p1). A total of

100 replicates of each phenotype were simulated using empirical genotypes of Pooled dataset. P-values were calculated using z test statistics (two sided).
(DOCX)

**S18 Table. SNP-based heritability of simulated traits from pooled dataset with correlated maternal-fetal genetic effects (average correlation = -0.5).** $\hat{h}^2$ of simulated traits from pooled dataset with correlated maternal-fetal genetic effects (average correlation = -0.5), estimated through conventional GCTA, M-GCTA and H-GCTA approach. Each approach was fitted using GREML ($\alpha$ = -0.25, -1.0), LDAK-Thin ($\alpha$ = -0.25, -1.0) and LDAK-Weights ($\alpha$ = -0.25, -1.0). For GCTA, M is the GRM generated from maternal genotypes (m), and F is the GRM generated from fetal genotypes (f). For M-GCTA, M' represents the genetic relationship matrix of mothers; G represents genetic relationship matrix of children and D represents mother-child covariance matrix. For H-GCTA, M1 is the GRM generated from maternal transmitted alleles (m1), M2 is the GRM generated from maternal non-transmitted alleles (m2), and P1 is the GRM generated from paternal transmitted alleles (p1). A total of 100 replicates of each phenotype were simulated using empirical genotypes of Pooled dataset. P-values were calculated using z test statistics (two sided).
(DOCX)

**S19 Table. SNP-based heritability of simulated traits from pooled dataset with correlated maternal-fetal genetic effects (average correlation = 1.0).** $\hat{h}^2$ of simulated traits from pooled dataset with correlated maternal-fetal genetic effects (average correlation = 1.0), estimated through conventional GCTA, M-GCTA and H-GCTA approach. Each approach was fitted using GREML ($\alpha$ = -0.25, -1.0), LDAK-Thin ($\alpha$ = -0.25, -1.0) and LDAK-Weights ($\alpha$ = -0.25, -1.0). For GCTA, M is the GRM generated from maternal genotypes (m), and F is the GRM generated from fetal genotypes (f). For M-GCTA, M' represents the genetic relationship matrix of mothers; G represents genetic relationship matrix of children and D represents mother-child covariance matrix. For H-GCTA, M1 is the GRM generated from maternal transmitted alleles (m1), M2 is the GRM generated from maternal non-transmitted alleles (m2), and P1 is the GRM generated from paternal transmitted alleles (p1). A total of 100 replicates of each phenotype were simulated using empirical genotypes of Pooled dataset. P-values were calculated using z test statistics (two sided).
(DOCX)

**S20 Table. SNP-based heritability of simulated traits from pooled dataset with correlated maternal-fetal genetic effects (average correlation = 0.5).** $\hat{h}^2$ of simulated traits from pooled dataset with correlated maternal-fetal genetic effects (average correlation = 0.5), estimated through conventional GCTA, M-GCTA and H-GCTA approach. Each approach was fitted using GREML ($\alpha$ = -0.25, -1.0), LDAK-Thin ($\alpha$ = -0.25, -1.0) and LDAK-Weights ($\alpha$ = -0.25, -1.0). For GCTA, M is the GRM generated from maternal genotypes (m), and F is the GRM generated from fetal genotypes (f). For M-GCTA, M' represents the genetic relationship matrix of mothers; G represents genetic relationship matrix of children and D represents mother-child covariance matrix. For H-GCTA, M1 is the GRM generated from maternal transmitted alleles (m1), M2 is the GRM generated from maternal non-transmitted alleles (m2), and P1 is the GRM generated from paternal transmitted alleles (p1). A total of 100 replicates of each phenotype were simulated using empirical genotypes of Pooled dataset. P-values were calculated using z test statistics (two sided).
(DOCX)

**S21 Table. SNP-based heritability of simulated fetal traits with parent-of-origin effects (POEs) from pooled dataset.** $\hat{h}^2$ of simulated fetal traits with POEs from pooled dataset,

estimated through conventional GCTA, M-GCTA and H-GCTA approach using GREML ($\alpha$ = -1.0) model – A) 50% causal variants with POEs; B) 25% causal variants with POEs. POEs were incorporated by reducing the effect of m1 as compared to p1 by multiplying effects of m1 with (1 – I) where I is the imprinting factor such as 0.25, 0.50, 0.75 and 1.0. In each scenario, m1 shows either no imprinting, i.e., I = 0.0 ( m1 / p1 = 1.0 / 1.0 ) or partial imprinting, i.e., I = 0.25-0.75 ( m1 / p1 = 0.75 / 1.0 − 0.25 / 1.0 ) or complete imprinting, i.e., I = 1.0 ( m1 / p1 = 0.0 / 1.0 ). For GCTA, M is the GRM generated from maternal genotypes (m), and F is the GRM generated from fetal genotypes (f). For M-GCTA, M' represents the genetic relationship matrix of mothers; G represents genetic relationship matrix of children and D represents mother-child covariance matrix. For H-GCTA, M1 is the GRM generated from maternal transmitted alleles (m1), M2 is the GRM generated from maternal non-transmitted alleles (m2), and P1 is the GRM generated from paternal transmitted alleles (p1). A total of 100 replicates of each phenotype were simulated using empirical genotypes of Pooled dataset. P-values were calculated using z test statistics (two sided).
(DOCX)

**S22 Table. SNP-based heritability of simulated traits from pooled dataset with correlated maternal-fetal genetic effects and parent-of-origin effects (POEs).** $\hat{h}^2$ of simulated traits from pooled dataset with correlated maternal-fetal genetic effects and POEs, estimated through conventional GCTA, M-GCTA and H-GCTA approach using GREML ($\alpha$ = -1.0) model. For simplicity, we assumed that all causal variants exhibit POEs. POEs were incorporated by reducing the effect of m1 as compared to p1 by multiplying effects of m1 with (1 – I) where I is the imprinting factor such as 0.25, 0.50, 0.75 and 1.0. m1 shows either no imprinting, i.e., I = 0.0 ( m1 / p1 = 1.0 / 1.0 ) or partial imprinting, i.e., I = 0.25-0.75 ( m1 / p1 = 0.75 / 1.0 − 0.25 / 1.0 ) or complete imprinting, i.e., I = 1.0 ( m1 / p1 = 0.0 / 1.0 ). For GCTA, M is the GRM generated from maternal genotypes (m), and F is the GRM generated from fetal genotypes (f). For M-GCTA, M' represents the genetic relationship matrix of mothers; G represents genetic relationship matrix of children and D represents mother-child covariance matrix. For H-GCTA, M1 is the GRM generated from maternal transmitted alleles (m1), M2 is the GRM generated from maternal non-transmitted alleles (m2), and P1 is the GRM generated from paternal transmitted alleles (p1). A total of 100 replicates of each phenotype were simulated using empirical genotypes of Pooled dataset. P-values were calculated using z test statistics (two sided).
(DOCX)

**S23 Table. SNP-based heritability of gestational duration and fetal size measurements at birth using all polymorphic SNPs.** Comparison of $\hat{h}^2$ estimated through conventional GCTA, M-GCTA and H-GCTA approach for A) gestational duration, B) birth weight, C) birth length and D) head circumference. Each approach was fitted using GREML ($\alpha$ = -0.25), LDAK-Thin ($\alpha$ = -1.0) and LDAK-Weights ($\alpha$ = -1.0). For GCTA, M is the GRM generated from maternal genotypes (m), and F is the GRM generated from fetal genotypes (f). For M-GCTA, M' represents the genetic relationship matrix of mothers; G represents genetic relationship matrix of children and D represents mother-child covariance matrix. For H-GCTA, M1 is the GRM generated from maternal transmitted alleles (m1), M2 is the GRM generated from maternal non-transmitted alleles (m2), and P1 is the GRM generated from paternal transmitted alleles (p1). Gestational duration was adjusted for fetal sex and fetal size measurements at birth were additionally adjusted for gestational duration up to third orthogonal polynomial. Analyses using GCTA and M-GCTA approach were adjusted for 20 PCs and H-GCTA approach was adjusted for 30 PCs (10 PCs corresponding to m1, m2 and p1 each). P-values were calculated using z test statistics (one sided).
(DOCX)

**S24 Table. SNP-based heritability of gestational duration and fetal size measurements at birth using SNPs with MAF > 0.001.** Comparison of $\hat{h}^2$ estimated through conventional GCTA, M-GCTA and H-GCTA approach for A) gestational duration, B) birth weight, C) birth length and D) head circumference. Each approach was fitted using GREML ($\alpha$ = -0.25, -1.0), LDAK-Thin ($\alpha$ = -0.25, -1.0) and LDAK-Weights ($\alpha$ = -0.25, -1.0). For GCTA, M is the GRM generated from maternal genotypes (m), and F is the GRM generated from fetal genotypes (f). For M-GCTA, M' represents the genetic relationship matrix of mothers; G represents genetic relationship matrix of children and D represents mother-child covariance matrix. For H-GCTA, M1 is the GRM generated from maternal transmitted alleles (m1), M2 is the GRM generated from maternal non-transmitted alleles (m2), and P1 is the GRM generated from paternal transmitted alleles (p1). Gestational duration was adjusted for fetal sex and fetal size measurements at birth were additionally adjusted for gestational duration up to third orthogonal polynomial. Analyses using GCTA and M-GCTA approach were adjusted for 20 PCs and H-GCTA approach was adjusted for 30 PCs (10 PCs corresponding to m1, m2 and p1 each). P-values were calculated using z test statistics (one sided).
(DOCX)

**S25 Table. SNP-based heritability of gestational duration and fetal size measurements at birth using SNPs with MAF > 0.01.** Comparison of $\hat{h}^2$ estimated through conventional GCTA, M-GCTA and H-GCTA approach for A) gestational duration, B) birth weight, C) birth length and D) head circumference. Each approach was fitted using GREML ($\alpha$ = -0.25, -1.0), LDAK-Thin ($\alpha$ = -0.25, -1.0) and LDAK-Weights ($\alpha$ = -0.25, -1.0). For GCTA, M is the GRM generated from maternal genotypes (m), and F is the GRM generated from fetal genotypes (f). For M-GCTA, M' represents the genetic relationship matrix of mothers; G represents genetic relationship matrix of children and D represents mother-child covariance matrix. For H-GCTA, M1 is the GRM generated from maternal transmitted alleles (m1), M2 is the GRM generated from maternal non-transmitted alleles (m2), and P1 is the GRM generated from paternal transmitted alleles (p1). Gestational duration was adjusted for fetal sex and fetal size measurements at birth were additionally adjusted for gestational duration up to third orthogonal polynomial. Analyses using GCTA and M-GCTA approach were adjusted for 20 PCs and H-GCTA approach was adjusted for 30 PCs (10 PCs corresponding to m1, m2 and p1 each). P-values were calculated using z test statistics (one sided).
(DOCX)

**S26 Table. SNP-based heritability of gestational duration and fetal size measurements at birth using SNPs with MAF > 0.05.** Comparison of $\hat{h}^2$ estimated through conventional GCTA, M-GCTA and H-GCTA approach for A) gestational duration, B) birth weight, C) birth length and D) head circumference. Each approach was fitted using GREML ($\alpha$ = -0.25, -1.0), LDAK-Thin ($\alpha$ = -0.25, -1.0) and LDAK-Weights ($\alpha$ = -0.25, -1.0). For GCTA, M is the GRM generated from maternal genotypes (m), and F is the GRM generated from fetal genotypes (f). For M-GCTA, M' represents the genetic relationship matrix of mothers; G represents genetic relationship matrix of children and D represents mother-child covariance matrix. For H-GCTA, M1 is the GRM generated from maternal transmitted alleles (m1), M2 is the GRM generated from maternal non-transmitted alleles (m2), and P1 is the GRM generated from paternal transmitted alleles (p1). Gestational duration was adjusted for fetal sex and fetal size measurements at birth were additionally adjusted for gestational duration up to third orthogonal polynomial. Analyses using GCTA and M-GCTA approach were adjusted for 20 PCs and H-GCTA approach was adjusted for 30 PCs (10 PCs corresponding to m1, m2 and p1 each). P-values were calculated using z test statistics (two sided).
(DOCX)

**S27 Table. Replication of heritability estimation of gestational duration.** $\hat{h}^2$ of gestational duration in HARVEST dataset based on SNPs with MAF > 0.01 estimated through H-GCTA using GREML (α = -1.0) model. Gestational duration was adjusted for fetal sex. P-values were calculated using z test statistics (one sided).
(DOCX)

**S1 Fig. Framework of the study.** Framework of the study depicting the traits under study, available datasets, MAF cutoffs, list of GRMs created in each MAF cutoff category, selection of unrelated mother-child pairs. Last block shows methods/models utilized for estimation and comparison of $\hat{h}^2$ estimated from our approach (H-GCTA) with those obtained by two available approaches – GCTA and M-GCTA.
(PDF)

**S2 Fig. Distribution of available phenotypes in datasets.** Distribution of available phenotypes in each dataset categorized by fetal sex – A) distribution of gestational duration, birth weight, birth length and head circumference in ALSPAC dataset; B) distribution of gestational duration, birth weight, birth length and head circumference in HAPO dataset; C) distribution of gestational duration, birth weight and birth length in FIN dataset; D) distribution of gestational duration and birth weight in DNBC dataset and E) distribution of gestational duration in MoBa dataset.
(PDF)

**S3 Fig. Comparison of $\hat{h}^2$ for simulated traits from ALSPAC dataset – maternal traits, fetal traits and traits with independent maternal-fetal genetic effects.** Comparison of $\hat{h}^2$ for simulated traits from ALSPAC dataset, estimated through different approaches fitting GREML (α = -1.0): A) maternal traits; B) fetal traits; C) traits where independent sets of causal variants have effects through mother and fetus; D) traits where same set of causal variants have effects through mother and fetus. For GCTA, M is the GRM generated from maternal genotypes (m), and F is the GRM generated from fetal genotypes (f). For M-GCTA, M' represents the genetic relationship matrix of mothers; G represents genetic relationship matrix of children and D represents mother-child covariance matrix. For H-GCTA, M1 is the GRM generated from maternal transmitted alleles (m1), M2 is the GRM generated from maternal non-transmitted alleles (m2), and P1 is the GRM generated from paternal transmitted alleles (p1). A total of 100 replicates of each phenotype were simulated using empirical genotypes of ALSPAC dataset. P-values were calculated using z test statistics (two sided). * = (p value <5.0E-02), ** = (p value <1.0E-02), *** = (p value <1.0E-03) and **** = (p value <1.0E-04).
(PDF)

**S4 Fig. Comparison of $\hat{h}^2$ for simulated traits from ALSPAC dataset –traits with correlated maternal-fetal genetic effects.** Comparison of $\hat{h}^2$ estimated through different approaches fitting GREML (α = -1.0) for simulated traits with joint maternal–fetal effects from ALSPAC dataset: A) average correlation = -1.0; B) average correlation = -0.5; C) average correlation = 1.0; D) average correlation = 0.5. For GCTA, M is the GRM generated from maternal genotypes (m), and F is the GRM generated from fetal genotypes (f). For M-GCTA, M' represents the genetic relationship matrix of mothers; G represents genetic relationship matrix of children and D represents mother-child covariance matrix. For H-GCTA, M1 is the GRM generated from maternal transmitted alleles (m1), M2 is the GRM generated from maternal non-transmitted alleles (m2), and P1 is the GRM generated from paternal transmitted alleles (p1). A total of 100 replicates of each phenotype were simulated using empirical genotypes of ALSPAC dataset. P-values were calculated using z test statistics (two sided). *

= (p value <5.0E-02), ** = (p value <1.0E-02), *** = (p value <1.0E-03) and **** = (p value <1.0E-04).
(PDF)

**S5 Fig. Schematic representation of variance attributable to maternal transmitted, maternal non-transmitted and paternal transmitted haplotypes in H-GCTA.** Schematic representation of variance attributable to maternal transmitted ( $\sigma_{m1}^2$ ), maternal non-transmitted ( $\sigma_{m2}^2$ ) and paternal transmitted ( $\sigma_{p1}^2$ ) haplotypes in H-GCTA – Z and W are the sets of causal variants with maternal and fetal effects, respectively. S1 (Black squares), S2 (orange circles) and S3 (purple triangles) are sets of causal variants with explicit maternal effects, joint-maternal-fetal effects and explicit fetal effects such that S1 $\in$ Z, S2 = Z $\cap$ W and S3 $\in$ W. $u_m$ and $u_f$ are causal effects through mother and fetus and $p_m$ and $p_f$ are reference allele frequencies of a causal variant in mother and fetus, respectively. Since each allele is a random draw from Bernoulli distribution, variance in terms of allele frequency is represented as $p_m(1-p_m)$ and $p_f(1-p_f)$ in mother and fetus, respectively. m1 affects the phenotype through maternal transmitted alleles in mother ( $g_{m1'}$ ) and maternal transmitted alleles in fetus ( $g_{m1''}$ ). Likewise, $g_{m2}$ and $g_{p1}$ represent the maternal non-transmitted and paternal transmitted alleles. Therefore, allelic effects - $u_{m1'} = u_{m2}$ , $u_{m1''} = u_{p1}$ (in the absence of POEs) and $cov(g_{m1'}, g_{m1''}) = \rho(g_{m1'}, g_{m1''}) \sigma_{g_{m1'}} \sigma_{g_{m1''}}$ is the covariance of two binomial random variables m1' and m1" present in mother and fetus, respectively; where, $\rho$ and $\sigma$ represent correlation and standard deviation of respective alleles. For a causal variant with joint maternal-fetal effect, $p_m = p_f = p$ in a random mating population, therefore, $cov(g_{m1'}, g_{m1''}) = p(1-p)$ and total phenotypic variance explained by m1, m2 and p1 is $2p(1-p)(u_m^2 + u_f^2 + u_m u_f)$.
(PDF)

**S6 Fig. Principal Components Analysis (PCA) plots using all polymorphic SNPs.** Principal Components Analysis (PCA) plots of unrelated (relatedness coefficient < 0.5) mother-child pairs using all polymorphic SNPs. m: Mothers' Genotypes; f: children's genotypes; m1: maternal transmitted alleles; m2: maternal non-transmitted alleles; and p1: paternal trans-mitted alleles. Unlike PCA using pooled data, 20 PCs were created using independent SNPs from a merged dataset which included SNPs from 1000 genome samples (phase 3) and pooled dataset. Since 1000 genome dataset in general lacks parent-child information, we used the first allele of phased 1000 genome data along with m1 or p1 to create 20 PCs whereas second allele of phased 1000 genome data was used along with m2 to create 20 PCs.
(PDF)

**S7 Fig. Replication of heritability estimation of gestational duration.** $\hat{h}^2$ estimation of fetal sex adjusted gestational duration in HARVEST dataset using our approach (H-GCTA). Estimated $\hat{h}^2$ of fetal sex adjusted gestational duration in pooled dataset is pasted for com-parison (image on the right). Analysis was performed through GREML (α = -1.0) using SNPs with MAF > 0.01.
(PDF)

## Acknowledgements

We are extremely grateful to all the families who participated in Avon Longitudinal Study of Parents And Children (ALSPAC), Hyperglycemia and Adverse Pregnancy Outcome study (HAPO), Finnish Birth Cohort (FIN), Danish Birth Cohort (DNBC) and Norwegian Mother, Father and Child Cohort study (MoBa), the clinical staff for their consistent help, the whole team of respective studies including interviewers, computer and laboratory technicians, cleri-cal workers, research scientists, volunteers, managers, receptionists and nurses. We also thank

organizing bodies for administrating the studies. Our sincere thanks to dbGaP for depositing and hosting data access for the current research. We thank the Norwegian Institute of Public Health (NIPH) for generating high-quality genomic data. We also thank the NORMENT Centre for providing genotype data, funded by the Research Council of Norway (#223273), South East Norway Health Authority and KG Jebsen Stiftelsen. We further thank the Center for Diabetes Research, the University of Bergen for providing genotype data and performing quality control and imputation of the data funded by the ERC AdG project SELECTion-PREDISPOSED, Stiftelsen Kristian Gerhard Jebsen, Trond Mohn Foundation, the Research Council of Norway, the Novo Nordisk Foundation, the University of Bergen, and the Western Norway health Authorities (Helse Vest).We also want to inform you that one of our co-authors, Dr. Kari Teramo is deceased now. We sincerely thank him for his contributions to our current research work.

## Author contributions

**Conceptualization:** Amit K. Srivastava, Ge Zhang.

**Data curation:** Amit K. Srivastava, Julius Juodakis, Pol Sole-Navais, Jonas Bacelis, Kari Teramo, Mikko Hallman, Pal R. Njolstad, Bo Jacobsson, Louis J. Muglia, Ge Zhang.

**Formal analysis:** Amit K. Srivastava, Julius Juodakis, Ge Zhang.

**Funding acquisition:** Ge Zhang.

**Investigation:** Amit K. Srivastava, Julius Juodakis, Pol Sole-Navais, Jing Chen, David M. Evans, Louis J. Muglia, Ge Zhang.

**Methodology:** Amit K. Srivastava, Ge Zhang.

**Project administration:** Ge Zhang.

**Resources:** Amit K. Srivastava.

**Supervision:** Louis J. Muglia, Ge Zhang.

**Validation:** Amit K. Srivastava, Julius Juodakis, Pol Sole-Navais, Ge Zhang.

**Visualization:** Amit K. Srivastava, Julius Juodakis, Jing Chen, Ge Zhang.

**Writing – original draft:** Amit K. Srivastava.

**Writing – review & editing:** Amit K. Srivastava, Julius Juodakis, Pol Sole-Navais, Jing Chen, Jonas Bacelis, Kari Teramo, Mikko Hallman, Pal R. Njolstad, David M. Evans, Bo Jacobsson, Louis J. Muglia, Ge Zhang.

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
