## [Decision Letter · Decision Letter 0]

16 Sep 2024

Dear Dr Zhang,

Thank you very much for submitting your Research Article entitled 'Haplotype-based analysis distinguishes maternal-fetal genetic contribution to pregnancy-related outcomes' to PLOS Genetics. We sincerely apologize for the atypical length of review. 

The manuscript was fully evaluated at the editorial level and by independent peer reviewers. The reviewers appreciated the attention to an important problem, but raised some substantial concerns about the current manuscript. Based on the reviews, we will not be able to accept this version of the manuscript, but we would be willing to review a much-revised version. We cannot, of course, promise publication at that time.

If you decide to revise the manuscript for further consideration at PLOS Genetics, please aim to resubmit within the next 60 days, unless it will take extra time to address the concerns of the reviewers, in which case we would appreciate an expected resubmission date by email to plosgenetics@plos.org.

If present, accompanying reviewer attachments are included with this email; please notify the journal office if any appear to be missing. They will also be available for download from the link below. You can use this link to log into the system when you are ready to submit a revised version, having first consulted our Submission Checklist .

PLOS has incorporated Similarity Check , powered by iThenticate, into its journal-wide submission system in order to screen submitted content for originality before publication. Each PLOS journal undertakes screening on a proportion of submitted articles. You will be contacted if needed following the screening process.

To resubmit, log into your Editorial Manager account and select the option 'Revise Submission' in the 'Submissions Needing Revision' folder.

We are sorry that we cannot be more positive about your manuscript at this stage. Please do not hesitate to contact us if you have any concerns or questions.

Yours sincerely,

Heather J Cordell

Academic Editor

PLOS Genetics

Michael Epstein

Section Editor

PLOS Genetics

Reviewer's Responses to Questions

**Comments to the Authors:**

Reviewer #1: Srivastava and colleagues present H-GCTA, a new method for heritability analysis, that partitions SNP heritability based on contributions from maternal inherited, maternal not-inherited and paternal inherited alleles. They first demonstrate the method on simulated data, then applied to two real phenotypes (gestational duration and fetal size), using mother-child data from five cohorts.

In general, I have a positive view of the paper. I think the method is novel, the simulations are fair and extensive, and the real analysis will be on interest. I therefore have only minor comments. Please note that, having almost completed my review, I realised I had reviewed this paper before (back in 2020), so have added some extra brief comments at the bottom.

#####

Minor comments

1 - I notice you restrict estimates to be non-negative. Personally, I prefer to not restrict, because restricting introduces biases. However, I doubt it has a big impact here, and can help when you have correlated GRMs.

2 - I wonder if necessary to use so many SNPs (i.e., imputed data). In general, using imputed is better, because it will capture more variation (and more causal variants, instead of variants tagging causal variants). However, doing so will lower precision relative to using just genotyped SNPs (both due to extra number of snps, and perhaps because imputed snps usually have more errors). Therefore, when sample size is not huge, you can find benefits using fewer snps (i.e., only directly genotyped, or even raising the MAF threshold to 0.05). However, this is just an idea, not a requirement.

3 - On a similar note, for your simulations, you could increase precision by considering a reduced genome (e.g., only chr 1-5). But as with 1, just an idea, not a requirement.

4 - L180 - I would suggest rewording "Although, negative values of ĥ2 usually mean nothing and are considered as zero, they are important for interpretation of results for traits with negative correlation of maternal-fetal genetic effects." Maybe say "negative values of h2 usually correspond to noise". Also, I was a bit confused by this sentence, given that methods say you do restrict estimate to be non-negative.

5 - I did not notice a software link, suggesting you used existing software. Therefore, I wondered how you created GRM from haplotype data? I think the easiest would be to save in bed format, where 0s and 1s are stored as 0 and 2 (doubling the values ensures LDAK / GCTA will standardize as per your formulae). It might be helpful for readers to explain how you performed the analyses. Please note that if you do not have bespoke software, and implementing is non-trivial, I am happy to add the feature to LDAK (i.e., construction of GRMs from haplotypes, outputed by Shapeit??)

6 - I appreciate Figure 1. However, I wonder if the word "interface" gives impression you consider interactions between mother and father?

7 - The simulation results figures. I am unsure whether you should highlight significance (wrt 0)? This is interesting for some comparisons (eg., where a method wrongly attributes variance). But in general, we are more interested in accuracy (closeness to true values)

8 - Figure 4 - Please ensure good resolution and clear labels (maybe the journal's conversion to pdf has damaged it). As p1 value is contact, can the x-axis simply be m1?

9 - The PC plot in Supp Figure 6 is a bit concerning, although partly because no reference genomes (e.g., you could overlay on 1000GP, so we can see the observed variation is small-scale, or global). However, while global variation might lead to variation, I believe it is unlikely to affect the relative values, which is the point of this study.

10 - Supp Figure 7, would help to add your primary results (so can compare with those from replication data)

11 - Naturally, it would be interesting to see results from a non pregnancy trait (e.g. height) as comparison / a control.

######

As noted, I eventually realised I previously reviewed this paper in 2020. I see that while generally positive, I criticised the use of only the "GCTA Model". Therefore, it greatly pleases me the authors listened to that criticism, and considered two alternative models. Thank you.

######

Signed Doug Speed

Reviewer #2: Srivastava et al. introduce an approach (H-GCTA) to partition the phenotypic variance of pregnancy outcomes into maternal transmitted, non-transmitted, and paternal transmitted alleles in mother-child pairs. Through simulation studies and real data analyses, the authors demonstrate that, compared to the M-GCTA method, the new H-GCTA can detect the contribution of parent-of-origin effects (POEs).

My major comment concerns the significance of the study and the simulation design.

- The authors did not provide a publicly available tool for the approach, which limits its potential use.

- Regarding the simulation study:

1) Why was the contribution of POEs studied only for traits only with fetal effects? Can we simulate data with different levels of POEs also for traits with varying correlations between maternal and fetal genetic effects? This would help in understanding more complex traits that involve both correlated maternal and fetal genetic effects and POEs.

2) It would be beneficial to calculate the true heritability attributable to maternal transmitted, non-transmitted, and paternal transmitted alleles in the simulated data (since we can compute the variance of each component relative to the variance of y), and then compare the results from GREML, LDAK-Thin, and LDAK-Weights to these true values.

3) Many observations from the simulation studies can be derived mathematically. For example, given the assumed correlation between maternal and fetal genetic effects (\rho), one can explicitly derive how this correlation affects the contributions of m1, m2, and p1. As noted in line 203, when \rho = 1, the expected contributions of m1, m2, and p1 follow a 4:1:1 ratio; line 167 when \rho = 0, the expected contributions of m1, m2, and p1 follow a 2:1:1 ratio. These can be generalized to express the ratio as a function of (\rho). Similarly, given an assumed imprinting factor, one can derive how it affects the ratio.

Minor comments:

- Line 152: The term "independent maternal-fetal genetic effects" is not accurate. A correlation = 0 does not imply independence.

- Line 82: M-GCTA was introduced without citations.

- In Figures 2, 3, 4, and 5, the y-axis labels are too densely packed, making it difficult to read the exact values.

**Have all data underlying the figures and results presented in the manuscript been provided?**

Reviewer #1: Yes

Reviewer #2: Yes

PLOS authors have the option to publish the peer review history of their article (what does this mean? ). If published, this will include your full peer review and any attached files.

**Do you want your identity to be public for this peer review?** For information about this choice, including consent withdrawal, please see our Privacy Policy .

Reviewer #1: **Yes: ** Doug Speed

Reviewer #2: No

---

## [Decision Letter · Decision Letter 1]

14 Jan 2025

Dear Dr Zhang,

We are pleased to inform you that your manuscript entitled "Haplotype-based analysis distinguishes maternal-fetal genetic contribution to pregnancy-related outcomes" has been editorially accepted for publication in PLOS Genetics. Congratulations!

Yours sincerely,

Heather J Cordell

Academic Editor

PLOS Genetics

Michael Epstein

Section Editor

PLOS Genetics

Aimée Dudley

Editor-in-Chief

PLOS Genetics

Anne Goriely

Editor-in-Chief

PLOS Genetics

Comments from the reviewers (if applicable):

Reviewer's Responses to Questions

**Comments to the Authors:**

Reviewer #1: Thank you for your responses, which answered all my questions. Also, it was nice to see the github scripts.

As a very small point, I looked at Supp Figure 9 (thank you for adding the 1000G populations as a reference). It was hard to see alspac, I guess because these points were plotted first, and the other populations plotted on top? Therefore, if easy, consider replotting ensuring the alspac points are on top (and perhaps in a unique colour)

Reviewer #2: The authors have significantly improved the manuscript by adding extensive simulations, clarifying the text, and providing well-organized example scripts for the analyses. I appreciate their efforts in addressing my concerns, and I believe all of my comments have been fully addressed.

**Have all data underlying the figures and results presented in the manuscript been provided?**

Reviewer #1: None

Reviewer #2: Yes

PLOS authors have the option to publish the peer review history of their article (what does this mean? ). If published, this will include your full peer review and any attached files.

**Do you want your identity to be public for this peer review?** For information about this choice, including consent withdrawal, please see our Privacy Policy .

Reviewer #1: **Yes: ** Doug Speed

Reviewer #2: No

**Data Deposition**

http://datadryad.org/submit?journalID=pgenetics&manu=PGENETICS-D-24-00393R1

**Press Queries**

---

## [Editor Report · Acceptance letter]

PGENETICS-D-24-00393R1

Haplotype-based analysis distinguishes maternal-fetal genetic contribution to pregnancy-related outcomes

Dear Dr Zhang,

We are pleased to inform you that your manuscript entitled "Haplotype-based analysis distinguishes maternal-fetal genetic contribution to pregnancy-related outcomes" has been formally accepted for publication in PLOS Genetics! Your manuscript is now with our production department and you will be notified of the publication date in due course.

With kind regards,

Anita Estes

PLOS Genetics

On behalf of:
